# Meta-analyses identify DNA methylation associated with kidney function and damage

Chronic kidney disease is a major public health burden. Elevated urinary albumin-to-creatinine ratio is a measure of kidney damage, and used to diagnose and stage chronic kidney disease. To extend the knowledge on regulatory mechanisms related to kidney function and disease, we conducted a blood-based epigenome-wide association study for estimated glomerular filtration rate ($n = 33{,}605$) and urinary albumin-to-creatinine ratio ($n = 15{,}068$) and detected 69 and seven CpG sites where DNA methylation was associated with the respective trait. The majority of these findings showed directionally consistent associations with the respective clinical outcomes chronic kidney disease and moderately increased albuminuria. Associations of DNA methylation with kidney function, such as CpGs at *JAZF1*, *PELI1* and *CHD2* were validated in kidney tissue. Methylation at *PHRF1*, *LDB2*, *CSRNP1* and *IRF5* indicated causal effects on kidney function. Enrichment analyses revealed pathways related to hemostasis and blood cell migration for estimated glomerular filtration rate, and immune cell activation and response for urinary albumin-to-creatinineratio-associated CpGs.

Chronic kidney disease (CKD) is a major public health burden. It affects more than 10% of adults worldwide and more than 40% of persons aged 70 years and older[1,2]. CKD is a leading cause of death worldwide[3], and is a major contributor to cardiovascular morbidity and mortality[2,4,5]. CKD is defined as the sustained presence of abnormalities of kidney structure or function. The kidney function measures most commonly used are the glomerular filtration rate, usually estimated from serum creatinine concentrations (eGFR), and the urinary albumin-to-creatinine ratio (UACR)[6].

Elevated UACR is a measure of kidney damage, used to diagnose and stage CKD[7], and is associated with diabetic and hypertensive kidney disease[8]. Even moderately elevated UACR is a risk factor for cardiovascular diseases, independently of other kidney function markers such as eGFR[9].

Familial aggregation studies of CKD and eGFR revealed a substantial heritable component of up to 54%[9–11]. Only a small part of this heritability is attributed to classical monogenic diseases. Rather, CKD susceptibility is influenced by DNA sequence variants in many genes, environmental factors, and their interactions. Genome-wide association studies (GWAS) have successfully identified common variants at >400 genetic loci that are associated with kidney function[10,12,13]. The index variants at known eGFR-associated loci explain an estimated 8.9% of eGFR variance[12].

A recent GWAS meta-analysis of eGFR integrated open chromatin regions with small sets of single nucleotide polymorphisms (SNPs)[10]. The results from this study support the importance of altered transcriptional regulation as a mechanism contributing to CKD. To investigate DNA methylation, with respect to kidney function, epigenome-wide association studies (EWAS) of eGFR and CKD have been carried out. As a key regulator of transcription that can be assessed in a cost-efficient and high-throughput manner, DNA methylation has been studied at CpG sites (CpGs) with single-base resolution. We previously conducted an EWAS including 4859 adults from two population-based studies and identified 18 validated, differentially methylated sites in whole blood associated with eGFR[14]. Although this study revealed insights into gene regulatory mechanisms of kidney function, the associated CpGs explained only 1.2% of the eGFR variance. Other previous studies were focused on CKD patients and/or patients with diabetes or patients with Human Immunodeficiency Virus infection, or were limited by small sample size, lack of replication, missing adjustment for potential confounders, or a combination thereof[14–19]. Other studies focused on DNA methylation patterns of diabetic kidney disease (DKD) patients[20–22].

Here, we conducted an EWAS of kidney function traits to identify additional CpGs related to gene regulatory mechanisms of potential importance to CKD. We extended the former EWAS for eGFR and CKD by substantially increasing the sample size to 33,605 individuals. Moreover, we included UACR and moderately increased albuminuria (microalbuminuria) as additional traits. The EWAS were conducted in predominantly population-based studies adjusting for sex, age, diabetes, hypertension, body mass index (BMI), smoking status, and the most abundant white blood cell proportions. We replicated our EWAS results in separate samples, related the CpG sites to gene expression levels in different tissues, applied the findings to clinical outcomes, and assessed causality between DNA methylation and kidney function (Supplementary Fig. 1).

## Results

**Study sample characteristics.** In this investigation, 36 studies with a total of 33,605 participants contributed to EWAS of eGFR and 15,068 to EWAS of UACR. Their pooled characteristics are shown in Table 1, and the individual study descriptions are provided in Supplementary Data 1 and 2.

**EWAS of eGFR and UACR.** We investigated the association of kidney traits with DNA methylation in blood at up to 441,870 CpGs, the overlap of CpGs covered by the Illumina MethylationEPIC BeadChip and the Illumina HumanMethylation450 BeadChip array, which were used for measurement by all but one study (Supplementary Data 3). All studies performed array data cleaning and applied centrally developed scripts for the preparation of the kidney trait values, which were subsequently related to DNA methylation using covariate-adjusted linear regression models with methylation β-values as the dependent variable following pre-specified study protocols (see Methods). We observed no or little inflation in study-specific EWAS (eGFR mean inflation = 1.00, UACR mean inflation = 1.00, Supplementary Data 1 and 2).

In a multivariable-adjusted trans-ethnic EWAS, 69 CpGs were significantly associated with eGFR and replicated (Fig. 1A, Supplementary Data 4, see Methods), including previously reported ones[14]. The replicated sites showed a clear pattern of lower methylation (60 CpGs, $p_{binom}$ = 2.2E−10; Fig. 1A).

**Table 1 Pooled characteristics of the discovery and replication samples.**

| EWAS Trait | eGFR | | UACR | |
|---|---|---|---|---|
| **EWAS stage** | **Discovery** | **Replication** | **Discovery** | **Replication** |
| Ancestries included | AA, EA, HIS, SA, SSA | AA, EA | AA, EA, HIS, SSA | AA, EA, AI |
| Sample size[a] | 22,318 | 11,359 | 11,579 | 3611 |
| Age, mean (SD) | 56.8 (12.5) | 56.0 (15.7) | 59.2 (12.2) | 58.0 (10.6) |
| Male, % (n) | 48.6 (10855) | 44.6 (5071) | 47.3 (5472) | 42.9 (1550) |
| Diabetes, % (n) | 12.6 (2813) | 8.3 (948) | 12.9 (1492) | 32.5 (1175) |
| Hypertension, % (n) | 44.3 (9888) | 48.2 (5480) | 49.5 (5729) | 49.0 (1770) |
| BMI (kg/m2), mean (SD) | 27.8 (5.2) | 27.6 (5.7) | 28.2 (5.3) | 30.3 (6.3) |
| Current smoking, % (n) | 14.5 (3236) | 16.1 (1832) | 13.3 (1538) | 28.6 (1032) |
| eGFR, mean (SD), mL/min/1.73m2 | 87.4 (19.4) | 91.2 (20.3) | 86.4 (19.5) | 93.9 (19.1) |
| UACR, median (1st, 3rd quartile), mg/g | NA | NA | 6.5 (4.0, 12.3) | 7.7 (3.8, 20.3) |
| CKD, % (n) | 7.8 (1741) | 7.4 (842) | NA | NA |
| microalbuminuria, % (n) | NA | NA | 10.4 (1207) | 18.4 (666) |

*AA* African American ancestry, *AI* American Indian ancestry, *EA* European ancestry, *HIS* Hispanics, *SA* South Asian ancestry, *SSA* Sub-Saharan African ancestry, *NA* not assessed, *EWAS* epigenome-wide association study, *SD* standard deviation, *eGFR* estimated glomerular filtration rate, *UACR* urinary albumin-to-creatinine ratio, *CKD* chronic kidney disease.
[a]The maximum sample size of the EWAS is lower due to missing methylation values for individual CpG sites.

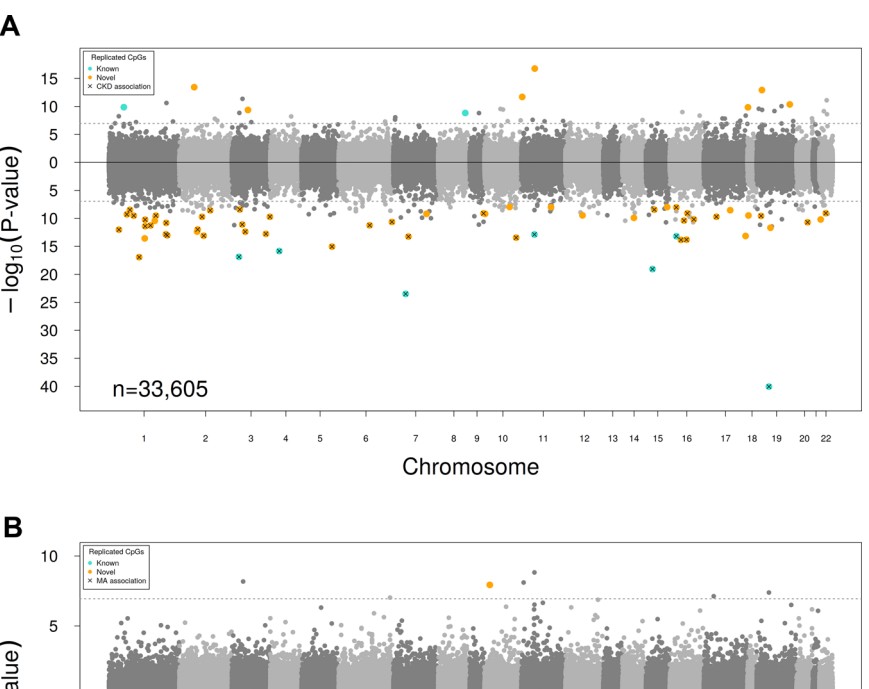

**Fig. 1 EWAS results of eGFR and UACR.** Chicago plots of the epigenome-wide association study (EWAS) results for estimated glomerular filtration rate (eGFR) (**A**) and urinary albumin-to-creatinine ratio (UACR) (**B**) using the combined discovery and replication sample. The sites are ordered by their chromosomal position on the x-axis, with their –log$_{10}$ P-value of the association Wald-test provided on the y-axis. CpGs positively correlated with the trait are plotted in the upper part, sites with negative correlation in the lower part. The dotted horizontal lines represent the level of significance (P-value < 1.1E−7). Novel replicates sites are colored in orange, known replicated sites are colored in turquois, and sites that were additionally associated with the respective binary trait (chronic kidney disease [CKD]/ microalbuminuria [MA]) are marked with a cross.

In the meta-analysis, the lowest P-values were observed for known eGFR associations including cg17944885 (*ZNF788*, β = −1.75E−04, P-value = 8.7E−41), cg23597162 (*JAZF1*, β = −1.48E−04, P-value = 3.2E−24), and cg06158227 (*ZSCAN29*, β = −1.08E−04, P-value = 8.7E−20)[14], followed by a new finding at cg20777437 (*CDCP2*, β = −8.28E−05, P-value = 1.1E−17). The 69 replicated eGFR-associated CpGs alone explained 15.7% of the eGFR variation in a separate study sample of 1888 participants (see Methods). When adding all covariates (sex, age, diabetes, hypertension, BMI, smoking, and white blood cell proportions) to the association model, the total proportion of explained variance in eGFR increased to 36.3%, with 2.4% variation attributed to the 69 CpGs independently of the other covariates.

In the trans-ethnic analysis of UACR, seven CpGs were significantly associated and replicated (Fig. 1B, Supplementary Data 5, see Methods). Of the findings, cg18181703 (*SOCS3*, β = −2.58E−03, P-value = 2.6E−13) and cg02711608 (*SLC1A5*, β = −1.63E−03, P-value = 9.9E−13) had the smallest meta-analysis association P-values. At six of the seven CpGs, lower methylation was associated with higher UACR. Due to the number of replicated sites, the power of the binomial test was limited (6 CpGs, $p_{binom}$ = 0.13; Fig. 1B). The replicated CpGs explained 3.9% and the full model 14.6% of variation in UACR

levels, with 0.07% attributed to the CpGs independently of the other covariates.

There was a low overlap between the replicated EWAS and previously reported GWAS loci for both eGFR ($n_{overlap}$ = 10 out of 69; Supplementary Data 4) and UACR ($n_{overlap}$ = 1 out of 7; Supplementary Data 5).

There was no overlap of replicated CpGs between eGFR and UACR, indicating trait-specific DNA methylation profiles in blood. Even among the 967 and 270 suggestive associations (P-value < 1E−05) of a combined discovery and replication sample meta-analysis for eGFR and UACR, respectively, only 10 sites overlapped between both traits. The detailed results from these meta-analyses for all suggestive associations are provided in Supplementary Data 6 (eGFR) and 7 (UACR).

**Ancestry heterogeneity and robustness of the findings**. To assess whether the association results on eGFR and UACR might be driven by a specific ancestry, we performed ancestry-stratified EWAS meta-analysis of European ancestry (EA) and African American ancestry (AA) samples with results from multiple studies contributing to these two ethnicities. Comparison of the association results of the replicated CpGs in samples of EA ($n_{eGFR}$ = 23,671; $n_{UACR}$ = 9806) with the AA samples ($n_{eGFR}$ =

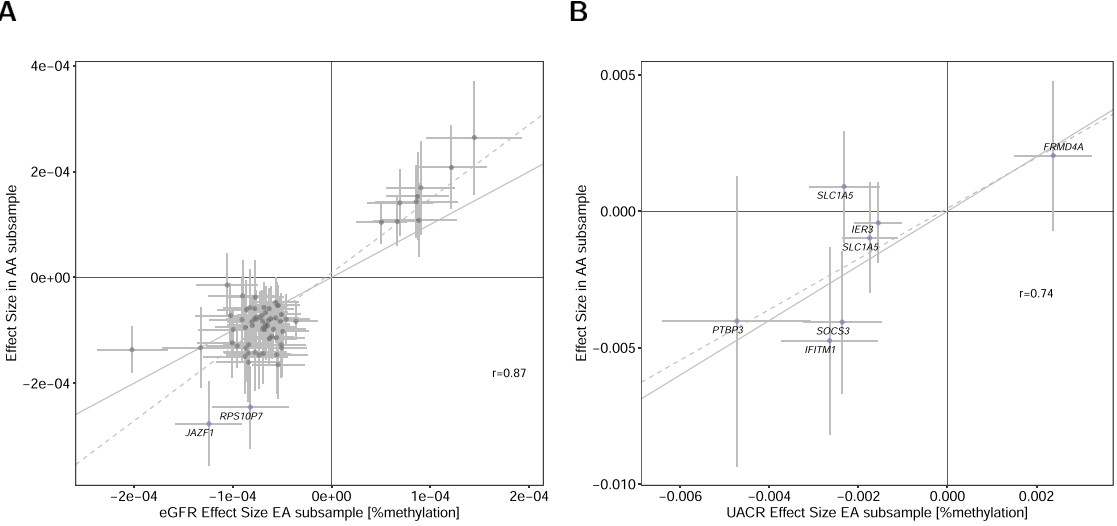

**Fig. 2 Similar effects between EA and AA-specific analyses.** Comparison of effect estimates of the association Wald-test of the significantly associated CpG sites (CpGs) for estimated glomerular filtration rate (eGFR) (**A**) and urinary albumin-to-creatinine ratio (UACR) (**B**). The effects in the European ancestry (EA) meta-analysis epigenome-wide association study (x-axis) ($n_{eGFR} = 23{,}671$; $n_{UACR} = 9806$) are compared with the corresponding effect sizes of the African American ancestry (AA) subsample (y-axis) ($n_{eGFR} = 5019$; $n_{UACR} = 1921$). In panel (**A**), sites that showed a significant P-value of the two-sided t-test for ancestry heterogeneity (P-value < 0.05/69) are colored in blue and labeled with the closest gene name. In all panels, the dashed gray line represents the linear regression slope between the dots, the solid gray line shows the diagonal, error bars indicate 95% CIs, and the Pearson correlation coefficient r between the effect estimates is shown.

5019; $n_{UACR} = 1921$) showed similar effect sizes ($r_{eGFR} = 0.87$, $r_{UACR} = 0.74$). The effect directions of almost all replicated CpGs were concordant between the ancestries (Fig. 2, Supplementary Data 4 and 5). The only exception was the UACR association cg22304262 in *SLC1A5* which, however, was not significant in AA (P-value = 0.39, Supplementary Fig. 2). This association, as well as the CpGs at *JAZF1* and *RPS10P7* for eGFR, showed significant heterogeneity (eGFR: P-value < 0.05/69, UACR: P-value < 0.05/7) between EA and AA association results (Fig. 2, Supplementary Data 4 and 5).

The presence of common SNPs (minor allele frequency >0.05) in or within 50 bp of the 69 replicated CpGs was assessed to evaluate whether the presence of SNPs could affect probe binding. Four probes were located near a common SNP (Supplementary Data 4; cg10960375-rs113564504; cg11544657-rs2083577; cg01817897-rs142643977; cg06930757-rs112223111), which, however, were not associated with any GWAS trait according to the PhenoScanner V2 resource (accessed 02/12/2021, $p_{GWAS} < 5E{-}8$, including proxy SNPs with EUR $r^2 > 0.8$)[23]. This indicates that the EWAS results are unlikely to be confounded by a common trait known to relate to kidney function by nearby DNA sequence variation. Probe-internal SNPs which were more than five bases from the 3′-end of the probe were generally found to be of negligible consequence[24].

**Relation to CKD and microalbuminuria.** Of the 69 CpGs associated with eGFR, 53 were also associated with prevalent CKD in a meta-analysis of 25,609 individuals including 2376 cases with a consistent effect direction (P-value < 0.05/69; Supplementary Data 4, Supplementary Fig. 3A). The correlation between eGFR and CKD effects was high ($r = -0.93$; Fig. 3A). In an independent cohort of 551 CKD patients, 65 CpGs were directionally consistent with the eGFR meta-analysis and five replicated (P-value < 0.05/69, $r = 0.77$; Supplementary Fig. 4, Supplementary Data 4, see Methods). In the same cohort, eGFR-associated CpGs cg18194850 (P-value = 1.6E−5) and cg07242931 (P-value = 2.3E−5) were also associated with time to kidney

failure or acute kidney injury (Supplementary Data 8, see Methods).

All seven CpGs associated with UACR were also significantly associated with microalbuminuria in a sample of 7279 individuals including 1186 cases with the same direction of effect as for UACR (P-value < 0.05/7, $r = 0.98$; Fig. 3B, Supplementary Data 5, Supplementary Fig. 3B).

**Correlation with gene expression.** To obtain insights into possible functional mechanisms of the eGFR- and UACR-associated CpGs, we tested the correlation of their DNA methylation levels with mRNA levels of genes encoded in *cis* in whole blood as well as in blood monocytes (see Methods, Supplementary Data 9). The known eGFR-associated cg17944885 on chromosome 19 near *ZNF788* was associated with the transcript encoded by the 240 kb distant zinc finger protein 439 (*ZNF439*, Table 2). In addition, methylation of cg04864179 at the interferon regulatory factor 5 (*IRF5*) was associated with both mRNA transcripts encoded by *IRF5* and the nearby transportin 3 (*TNPO3*) (Supplementary Fig. 5A).

Of the UACR associations, the two replicated CpGs cg02711608 and cg22304262 at solute carrier family 1 member 5 (*SLC1A5*) revealed significant associations with *SLC1A5* mRNA levels, and cg23570810 at the interferon induced transmembrane protein 1 (*IFITM1*) with its transcripts in *cis* (Table 2). All results of the gene expression analysis are provided in Supplementary Data 9.

**Effects in kidney tissue.** To assess whether the observed methylation effects in blood translate to kidney tissue, we applied a regression model to test the associations of the replicated CpGs for significant (false discovery rate (FDR) < 0.05) and direction-consistent association with eGFR and kidney fibrosis, respectively, in 506 microdissected kidney tissue samples.

In these samples, kidney-tissue-based DNA methylation of replicated eGFR-associated CpGs cg23597162 at *JAZF1*, cg26099045 near *PELI1*, and cg12644285 at *CHD2* were significantly associated with eGFR with the same effect direction as in blood (Supplementary Fig. 6A, Table 2, Supplementary Data 10). Furthermore, the same CpG at *PELI1* and six additional

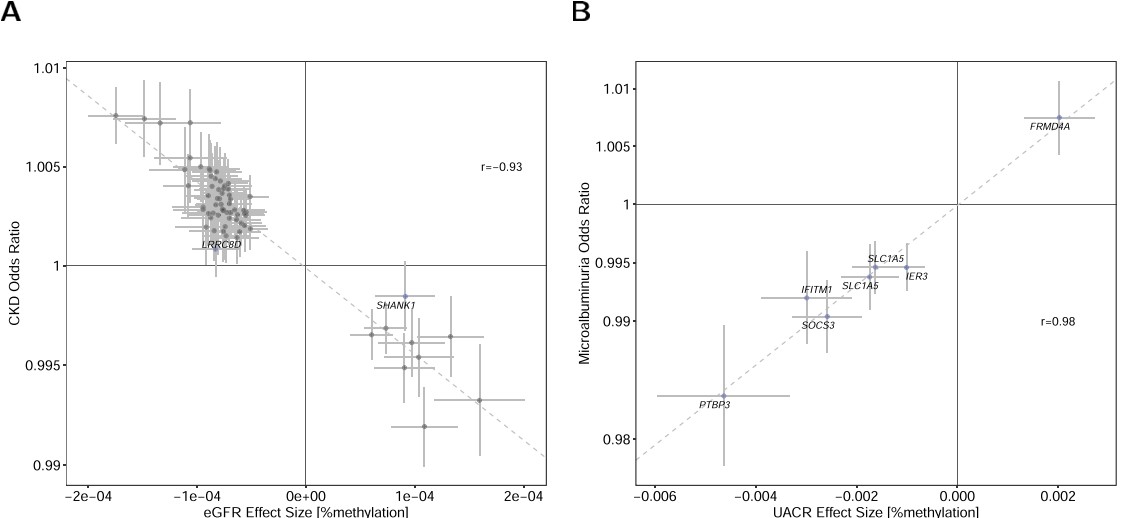

**Fig. 3 Effects of eGFR-associated sites on CKD and of UACR-associated sites on Microalbuminuria.** Comparison of effect estimates of the association Wald-test of the significantly associated CpG sites for estimated glomerular filtration rate (eGFR) ($n = 33,605$) (**A**) and urinary albumin-to-creatinine ratio (UACR) ($n = 15,068$) (**B**). The effects in the combined epigenome-wide association study (x-axis) are compared with the corresponding effect sizes (odds ratio) for chronic kidney disease (CKD) ($n = 25,609$) and microalbuminuria ($n = 7279$), respectively (y-axis). In panel (**A**), sites that were not nominally significantly associated with CKD (association test P-value≥0.05) are colored in blue and labeled with the closest gene name. In all panels, the dashed line represents the linear regression slope between the dots, error bars indicate 95% CIs, and the Pearson correlation coefficient r between the effect estimates is shown.

sites (cg20146909 at *LRRC8D*, cg25767870 near *FAM46C*, cg16618493 at *ZBTB7B*, cg10632966 near *RPEL1*, cg01068906 at *NOD2*, and cg03297731 at *GDPD3*) were associated with fibrosis in the kidney tissue samples with consistent (i.e., inverse) effect directions (Supplementary Fig. 6B, Table 2).

Of the replicated UACR sites, the DNA methylation levels in kidney tissue of cg00008629 at *PTBP3* and cg24859433 near *IER3* were associated with fibrosis and showed the same direction of effect (Supplementary Fig. 6D, Table 2). Consistent with the EWAS findings, no significant association of the UACR-associated CpGs with eGFR in the kidney sample donors was found (Supplementary Fig. 6C, Supplementary Data 10).

**Causal effects between DNA methylation and kidney function traits.** To assess whether the kidney function traits causally affect DNA methylation or vice versa, we conducted a bi-directional two-sample Mendelian randomization (MR) analysis of the significantly associated CpGs. The forward direction MR analysis suggested that the CpGs cg02304370 (*PHRF1*), cg04460609 (*LDB2*), cg00501876 (*CSRNP1*), and cg04864179 (*IRF5*) causally influence eGFR levels (Table 3). The heritability estimates of these four CpGs varied among the three datasets from populations of European ancestry, but were consistently higher than the mean heritability across all CpGs assessed in each of the three studies: 0.34 (causal eGFR CpGs) vs. 0.16 (HM450K array) based on Hannon et al.[25], 0.55 vs. 0.19 for Dongen et al.[26], and 0.51 vs. 0.19 for McRae et al.[27]. No significant causal associations were identified for any UACR-associated CpGs.

For the reverse MR, the only significant effect was that of eGFR on cg23597162 (*JAZF1*) when using the primary inverse-variance method. However, no significant effects were identified/observed in the multiple reverse MR sensitivity analyses. Leave-one-out analyses did not show any indication that MR results might be driven by a single SNP. Furthermore, excluding SNPs that were associated with type 2 diabetes mellitus being a major risk factor for kidney disease did not result in different findings.

Results for all primary and sensitivity MR analyses are shown in Supplementary Data 11 and 12 and Supplementary Fig. 7. The

sensitivity analyses support the primary findings of the significant forward MR results (FDR < 0.05, see Methods) with direction-consistent effect estimates, indicating potentially causal relationships from DNA methylation to eGFR but not vice versa.

**Transcription factor binding, histone mark and pathway enrichment analyses.** We performed transcription factor binding site, histone mark and pathway enrichment analyses based on the 967 CpGs that showed suggestive association with eGFR (P-value < 1E−05; Supplementary Data 6) and the 270 CpGs that showed suggestive associations with UACR (P-value < 1E−5; Supplementary Data 7) in the meta-analysis to maximize statistical power of enrichment analyses (see Methods)[14].

First, we evaluated whether eGFR-associated CpGs preferentially mapped to binding sites of 169 transcription factors (TFs) based on chromatin immunoprecipitation DNA-sequencing (ChIP-seq) data from the ENCODE project that was aggregated by a consensus calls across 91 human cell types (161 TF tracks) and supplemented with eight kidney tissue-based tracks (see Methods). After multiple testing correction for 169 TFs, eight TFs showed significant enrichment for eGFR-associated CpGs (FDR < 0.05; Fig. 4A, Supplementary Data 13), including CEBPB (P-value = 1.75E−06, cross-tissue track) and EP300 (P-value = 7.49E−09, cross-tissue track). This is consistent with the finding from Chu et al.[14] in a smaller subset of 4859 participants of the 33,605 participants analyzed here. UACR-associated CpGs were enriched in binding sites of 56 TFs (Fig. 4B, Supplementary Data 14) with the strongest association for POLR2A (P-value = 6.93E−19), FOS (P-value = 1.28E−12) and EP300 (P-value = 1.20E−10).

Kidney-function-associated CpGs were broadly enriched for several histone marks, with 82 of 195 histone mark cell type combinations significant (FDR q < 0.05) for eGFR (Fig. 4C, Supplementary Data 15) and 79 of 195 for UACR (Fig. 4D, Supplementary Data 16). The histone modification H3K4me1, a mark concentrated at active and primed enhancers, was enriched for all cell types for UACR-associated CpGs, and almost all cell types for eGFR-associated CpGs. Whereas UACR-associated CpGs showed enrichment for the promoter mark H3K4me3 in 34 cell types, only four cell

**Table 2 Meta-analysis results of the eGFR and UACR EWAS with functional support.**

| probeID | Chromosomal position (b37) | Nearest gene | N total EWAS | Effect size | Standard error | P-value (Wald-test) | CKD/MA effect direction | Association in kidney tissue | Correlation with mRNA expression | Other EWAS trait association |
|---|---|---|---|---|---|---|---|---|---|---|
| **eGFR** | | | | | | | | | | |
| cg17944885[a] | 19:12,225,735 | ZNF788 | 33,592 | −1.75E−04 | 1.31E−05 | 8.74E−41 | + | | ZNF439 | |
| cg23597162[a] | 7:28,102,341 | JAZF1 | 33,592 | −1.48E−04 | 1.46E−05 | 3.18E−24 | + | eGFR | | |
| cg20146909 | 1:90,289,611 | LRRC8D | 33,594 | −8.31E−05 | 1.09E−05 | 2.65E−14 | + | Fibrosis | | Smoking |
| cg26099045 | 2:64,291,800 | PELI1 | 33,595 | 1.59E−04 | 2.11E−05 | 3.65E−14 | − | eGFR, fibrosis | | Sex, smoking |
| cg25767870 | 1:118,188,756 | FAM46C | 33,595 | −7.27E−05 | 1.05E−05 | 5.37E−12 | + | Fibrosis | | |
| cg03297731 | 16:30,124,293 | GDPD3 | 33,578 | −7.04E−05 | 1.07E−05 | 4.12E−11 | + | Fibrosis | | |
| cg16618493 | 1:154,978,980 | ZBTB7B[b] | 32,064 | −5.64E−05 | 8.97E−06 | 3.15E−10 | + | Fibrosis | | |
| cg04864179 | 7:128,579,964 | IRF5[b] | 33,596 | −8.53E−05 | 1.38E−05 | 6.48E−10 | + | | IRF5, TNPO3 | |
| cg01068906 | 16:50,745,944 | NOD2 | 33,585 | −8.89E−05 | 1.45E−05 | 8.04E−10 | + | Fibrosis | | |
| cg12644285 | 15:93,570,953 | CHD2 | 33,603 | −8.07E−05 | 1.41E−05 | 1.04E−08 | + | eGFR | | CRP |
| cg10632966 | 10:105,001,051 | RPEL1[b] | 33,588 | −5.64E−05 | 9.86E−06 | 1.08E−08 | + | Fibrosis | | |
| **UACR** | | | | | | | | | | |
| cg02711608 | 19:47,287,964 | SLC1A5 | 14,505 | −1.63E−03 | 2.28E−04 | 9.85E−13 | − | | SLC1A5 | BP, alcohol, BMI, GGT[c] |
| cg00008629 | 9:115,093,661 | PTBP3 | 14,483 | −4.63E−03 | 6.69E−04 | 4.46E−12 | − | Fibrosis | | |
| cg23570810 | 11:315,102 | IFITM1 | 14,500 | −2.99E−03 | 4.60E−04 | 7.99E−11 | − | | IFITM1 | |
| cg22304262 | 19:47,287,778 | SLC1A5 | 14,486 | −1.74E−03 | 2.92E−04 | 2.56E−09 | − | | SLC1A5 | BP, GGT[c] |
| cg24859433 | 6:30,720,203 | IER3[b] | 14,485 | −1.01E−03 | 1.79E−04 | 1.84E−08 | − | Fibrosis | | Edu[c] |

BMI: body mass index; BP: systolic and diastolic blood pressure; CRP: C-reactive protein; CKD: chronic kidney disease; CRP: C-reactive protein; Edu: Educational attainment.
GGT: gamma-glutamyl transferase; EWAS: epigenome-wide association study; CKD: chronic kidney disease; MA: microalbuminuria.
[a]Known EWAS association with eGFR.
[b]Locus associated with GWAS of corresponding trait (±1 MB).
[c]Additional association with serum metabolites.

**Table 3 CpG sites having a potentially causal effect on eGFR as assessed by Mendelian randomization.**

| probeID | N instruments | Effect size | Standard error | P-value (Wald-test) | FDR | Nearest gene |
|---|---|---|---|---|---|---|
| cg02304370 | 6 | 4.22E−03 | 1.09E−03 | 1.11E−04 | 0.004 | PHRF1 |
| cg04460609 | 7 | 2.61E−03 | 7.63E−04 | 6.25E−04 | 0.015 | LDB2 |
| cg00501876 | 3 | 9.92E−03 | 3.00E−03 | 9.45E−04 | 0.019 | CSRNP1 |
| cg04864179 | 21 | 3.13E−03 | 9.62E−04 | 1.12E−03 | 0.020 | IRF5 |

Results of the inverse-variance weighted Mendelian randomization of the DNA methylation on estimated glomerular filtration rate (eGFR) levels. The effect estimates provide the per-unit change in one standard deviation of DNA methylation levels on natural log-transformed eGFR. FDR: false discovery rate.

types showed enrichment for eGFR-associated sites. Conversely, the gene body-associated mark H3K36me3 showed a much broader enrichment for eGFR-associated CpGs (30 cell types) than UACR-associated CpGs (five cell types).

As opposed to H3K3me1/3 and H3K36me3, which have all been linked to active genes (enhancers, active promotors, active transcription), H3K9me3 and H3K27me3, which are generally associated with constitutive and facultative heterochromatin[28–31], were not significantly enriched for UACR-associated CpGs in any cell type tested (Fig. 4D, Supplementary Data 16).

Enrichment of genes implicated by kidney-function-associated CpGs was assessed in the Gene Ontology (GO), the Kyoto Encyclopedia of Genes and Genomes (KEGG) and Reactome databases (see Methods)[32–35]. Significant enrichment was observed for 27 terms (25 GO, one KEGG, one Reactome; Supplementary Data 17, Fig. 5A) for eGFR, and 91 terms (87 GO, four Reactome; Supplementary Data 18, Fig. 5B) for UACR (Fig. 5B). Pathways related to white blood cell type specific migration, mRNA translation, hemostasis and coagulation, as well as insulin response showed significant enrichment for eGFR-associated CpGs (FDR < 0.05; Fig. 5A). The pathways with the lowest enrichment P-values for UACR-associated sites were dominated by immune cell activation and response, and interferon-related pathways (Fig. 5B). Additional pathways that were enriched included white blood cell migration, as for the eGFR results (Supplementary Data 18).

**Relation to known EWAS associations with other traits.** Given the observed enrichment of immune response-related pathways, the DNA methylation results including CpGs near genes of the interferon pathway, and the fact that the DNA methylation status was predominantly assessed in leukocytes, we assessed whether our findings may be driven by confounding effects of immune response or inflammation status. Thus, we performed a lookup of the replicated CpGs in a large EWAS on high-sensitive serum C-reactive protein (CRP) levels[36]. Only two CpGs, which were also associated with DNA methylation and eGFR in kidney tissue, cg09610644 at BDH1 and cg12644285 at CHD2, were among the CRP-associated sites. Thus, a general confounding of our results through inflammatory status as estimated by CRP seems unlikely.

Some of the kidney-function-associated CpGs identified in this study are associated with several other traits based on published EWAS. Twenty-one eGFR sites were associated with blood pressure, alcohol consumption, BMI, sex, soluble tumor necrosis factor receptor 2, and/or smoking status, and five UACR sites were previously associated with alcohol consumption, smoking status, BMI, educational attainment, γ-glutamyl transferase, soluble tumor necrosis factor receptor 2, type 2 diabetes mellitus, and/or several serum metabolites (Table 2, Supplementary Data 19). Three of these CpGs (all UACR sites) were associated with more than two traits, namely cg02711608 in SLC1A5 (with γ-glutamyl transferase, γ-glutamylthreonine, BMI, alcohol consumption), cg18181703 in SOCS3 (with type 2 diabetes mellitus,

smoking status, BMI, soluble tumor necrosis factor receptor 2), and cg24859433 near IER3 (with smoking status, educational attainment, 4-vinylphenol_sulfate).

Supplementary Fig. 5B exemplifies a CpG, cg26099045, that was correlated with eGFR and fibrosis in kidney tissue in our analysis, and additionally associated with sex and smoking in previous EWAS. Finally, cg17944885 at ZNF788 was associated also with eGFR in a sample of DKD patients[22].

## Discussion

In this EWAS of kidney function, we identified and replicated blood DNA methylation levels of 69 CpGs associated with eGFR. Of these, 60 sites were previously not reported, as were all seven CpGs identified in association with UACR. The majority of the eGFR- and UACR-associated CpGs were also significantly associated with their clinical outcomes, i.e., CKD and micro-albuminuria, and may therefore have potential for stratification of individuals at risk.

The variance of eGFR attributed to these 69 CpGs was 2.4%—twice that of an earlier EWAS[14], and is comparable to the variance explained by 29 SNPs discovered by GWAS that included three times higher sample size for locus discovery[37,38]. This suggests that differential DNA methylation at individual CpGs quantified from blood explains more of the eGFR variance than individual common SNPs at a given sample size. For UACR, it seems that larger sample sizes compared to eGFR are required to reveal a similar number of trait-associated CpGs. Given that albumin and creatinine for UACR calculation are measured in urine as opposed to the quantification of creatinine from serum for the estimation of GFR, this difference is not unexpected, and it is in line with observations from GWAS of these traits[10,13]. Furthermore, genetic variation seems to affect the kidney function traits via different pathways than changes in DNA methylation. This is supported by the low overlap between the replicated EWAS and GWAS sites for both eGFR and UACR.

This EWAS meta-analysis included samples of different populations and ethnicities. We observed a high correlation of the effect estimates obtained from the large subsample of EA individuals with the effects estimated in the AA individuals (Fig. 2). Despite the inclusion of data from a substantial number of individuals of non-EA ancestry, the overall results of the trans-ethnic EWAS might be driven by data from individuals of EA ancestry, which represented 70% (eGFR) / 65% (UACR) of our study (Supplementary Data 3 and 4). To address this limitation, future analyses with an increased proportion of non-EA samples are needed for a reliable and detailed assessment of between-ancestry heterogeneity.

The potential to translate insights from EWAS of quantitative traits in mostly population-based studies to persons with disease is illustrated by the fact that effect estimates at 65 of 69 eGFR-associated CpGs were observed in the same direction in a cohort of 551 CKD patients, pointing towards mechanisms applicable across a broad range of eGFR levels (Fig. 3). Moreover,

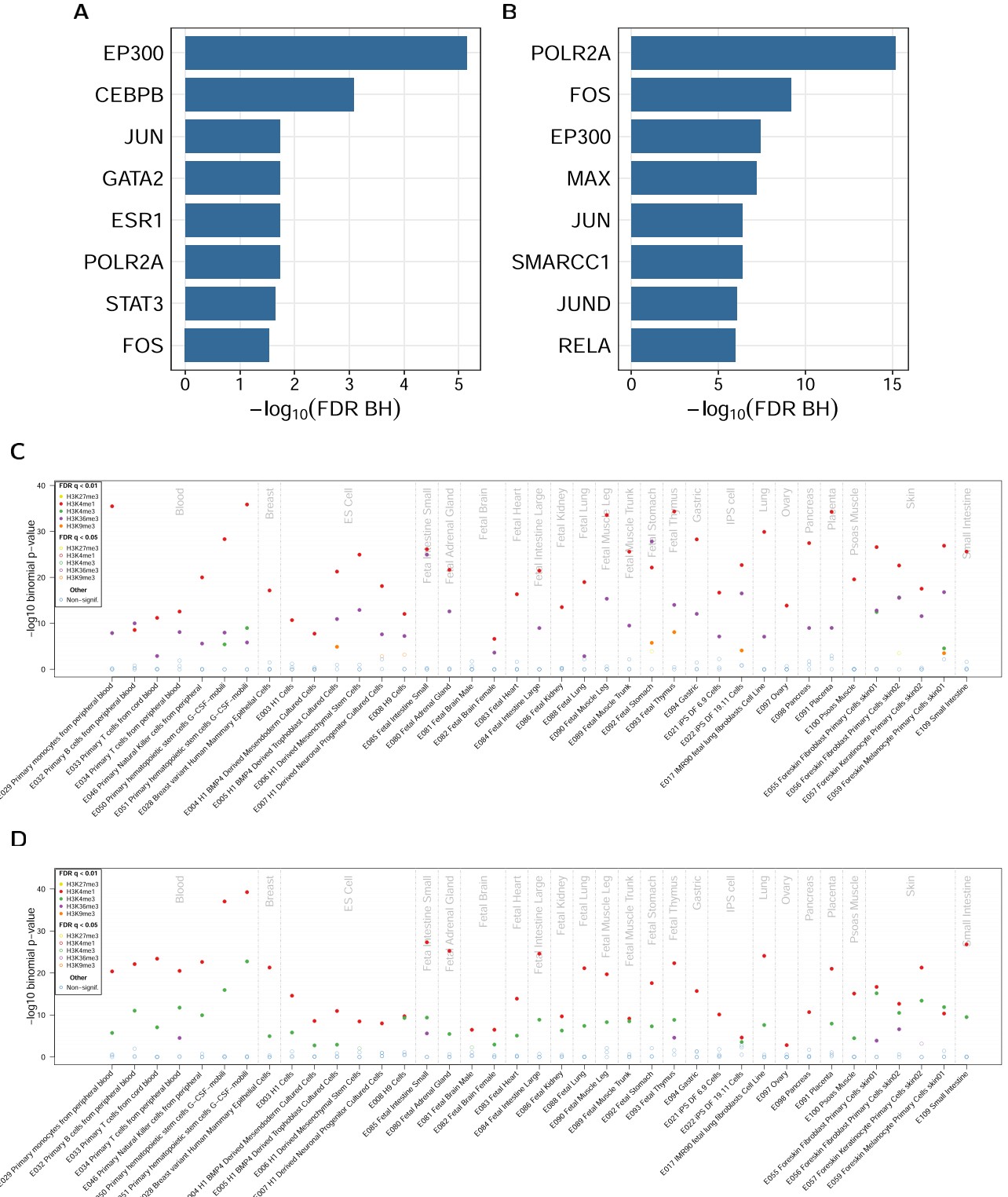

**Fig. 4 Enrichment in transcription factor binding sites and histone marks.** Enrichment analysis of CpG sites (CpGs) significantly associated with estimated glomerular filtration rate (eGFR) (**A**) and urinary albumin-to-creatinine ratio (UACR) (**B**) for mapping into regions containing specific transcription factor binding sites at Benjamini-Hochberg FDR < 0.05. Panels (**C**) and (**D**) show the corresponding results for mapping into histone marks. In these panels, the Y-axis show the $-\log_{10}(P\text{-value})$ from a binomial test comparing the expected and observed numbers of significant CpGs that map into the binding site regions for a given target. On the X-axis, the results with an FDR < 0.05 of 169 evaluated transcription factors are listed in alphabetical order. Enrichment testing was carried out using permutation with matching for genomic localization when sampling from the background.

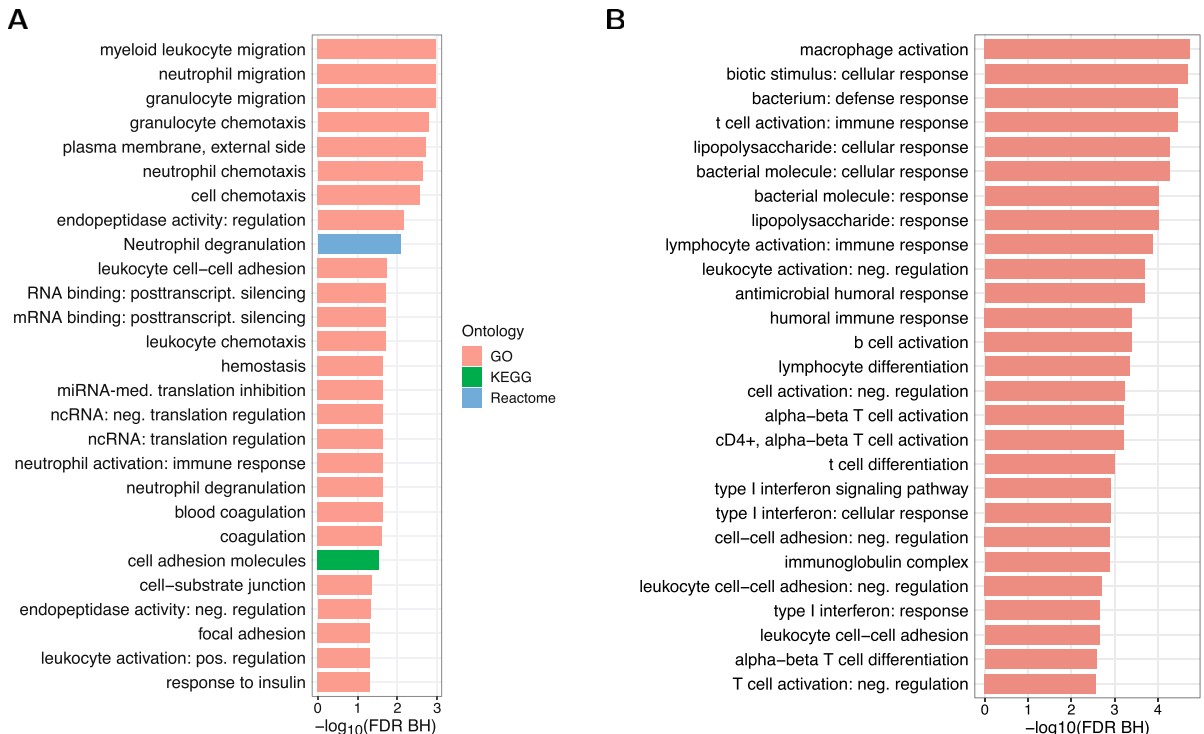

**Fig. 5 Pathway enrichment results.** The enrichment results of genes implicated by estimated glomerular filtration rate (eGFR) (**A**) and urinary albumin-to-creatinine ratio (UACR) (**B**) associated CpG sites as assessed in the Gene Ontology (GO), the Kyoto Encyclopedia of Genes and Genomes (KEGG) and Reactome databases are shown. The results passing a Benjamini-Hochberg FDR < 0.05 are shown but are limited to the top 27 pathways for UACR.

cg07242931 in *MAN1C1* and cg18194850 in *SUCLG2* were not only associated with eGFR, but predicted time to kidney failure or acute kidney injury (Supplementary Data 8). Further evidence for an involvement of *SUCLG2*, which encodes the β-subunit of succinyl-CoA synthetase, in kidney disease was suggested by a GWAS on diabetic kidney disease in American Indians (SNP = rs4453858, *P*-value = 2E−6)[39]. The genetic variation in this gene was also associated with urinary levels of succinyl-carnitine (C4DC, SNP = rs115560420, *P*-value = 7E−15)[40]. C4DC is a metabolic product from the tricarboxylic acid cycle, which is catalyzed under involvement of succinyl-CoA synthetase, and higher C4DC levels in blood associate with a lower eGFR[41]. However, a lookup in the methylation quantitative trait loci (meQTL) results of GoDMC (http://www.godmc.org.uk)[42] did not reveal a significant association of these two SNPs with cg18194850, suggesting no direct link between these SNPs and the CpG site.

As DNA methylation is a major regulator of gene expression, we assessed the correlation of the trait-associated DNA methylation sites with mRNA levels of genes in *cis*. Genes whose mRNA levels significantly correlated with kidney function associated DNA methylation sites were related to interferon pathways (both for eGFR and UACR). Although these findings agree with the pathway enrichment results for UACR (Fig. 5B, Supplementary Data 18), the number of significant correlations between UACR-associated CpGs and transcript levels is quite low. This is unsurprising considering the relatively small sample size of 1915 individuals available for this analysis and the limited coverage of the transcriptomics resource. Interestingly, among the significant findings for UACR, two CpGs at the solute carrier family 1 member 5 (*SLC1A5*) were associated with differential gene expression and also with blood pressure. DNA methylation at *SLC1A5*, which encodes a sodium-dependent neutral amino acid transporter, could therefore be an additional element contributing to the known correlation between UACR and blood pressure.

Significant differentially expressed transcripts were not always encoded by the closest gene (cg17944885 near *ZNF788*), or a CpG site was correlated with multiple genes in *cis* (cg04864179 at *IRF5*). In particular, the correlation of DNA methylation at cg04864179 with *IRF5* transcript levels is notable. Considering the significant MR result of this CpG with eGFR, DNA methylation at *IRF5* might causally influence eGFR mediated by its gene expression. Taking the trait transformation of the underlying genetic associations into account (see Methods), we observed that an increase in DNA methylation of ten standard deviations resulted in 3.2% higher eGFR. Although this estimated effect is very small, a causal influence of methylation patterns in blood cells on CKD was also supported by summary-based MR with a different meQTL dataset in a recent study, where the causal effect was directionally consistent with our results[22]. By applying multi-trait colocalization, they also showed that the genetic effects of this locus on eGFR were mediated by *IRF5* methylation and gene expression in blood. Furthermore, colocalization of *IRF5* gene expression with eGFR was shown in both tubular and glomerular compartments of kidney tissue[43]. When interpreting the causal effect observed in our study, it should however be kept in mind that the MR analyses for CpG cg04864179 showed significant heterogeneity (*p* < 0.05) among the instruments (Supplementary Data 11 and Supplementary Fig. 7).

*IRF5* encodes a member of the interferon regulatory factor (IRF) family. The group of IRF family members contains transcription factors with different roles, such as the modulation of immune system activity, growth and differentiation, as well as the control of gene expression for the interferon response to viral infections. *IRF5* can influence immune cell response. Multiple studies support that changes in *IRF5* methylation might affect kidney function via immune pathways: SNPs in *IRF5* are associated SLE via changes in *IRF5* expression in blood monocytes[44–46]. SLE is an autoimmune disease characterized by an activation of the IFN pathway that can affect the kidneys as

lupus nephritis[47]. Inhibition of *IRF5* hyperactivation in a mouse model of SLE protected from lupus nephritis onset and severity, and improved kidney function and pathology[48,49]. Although our EWAS did not focus on the study of SLE patients, small effects of *IRF5* methylation on kidney-related outcomes, mediated at least partially by its expression and subsequent IFN pathways, might be detected as effects on eGFR in the general population.

Several of the eGFR-associated CpGs for which DNA methylation was quantified from blood cells were also associated with eGFR and kidney fibrosis when DNA methylation was quantified from kidney tissue, although the sample size was substantially smaller compared to the EWAS dataset. This suggests that at least some of the findings obtained from blood can be translated to an additional trait-specific target tissue. Here, blood and kidney are the two main trait-specific target tissues, since the function of the kidney is filtration of blood to remove waste.

Enrichment analyses of the kidney function associated CpGs indicated a central role in transcriptional regulation. We found widespread enrichment of H3K4me1/3 and H3K36me3. H3K4me1/3 is linked to primed and active enhancers as well as active promoters and H3K36me3 is tightly correlated with transcribed regions of the genome[50]. The role of transcriptional regulation is further supported by the strongest transcription factor enrichment signal of UACR-associated CpGs, which is POLR2A, the largest subunit of the major enzyme synthesizing mRNA in eukaryotes.

Potential limitations related to the MR analyses include that valid instruments were not available for all CpGs for the forward MR. Given that all instruments of a CpG were selected from *cis* regions, i.e. of the same genetic region, it is likely that all instruments of a CpG are either valid or invalid, thus limiting the number of different MR methods that can be applied to test the robustness of the results[51]. The reverse MR was limited in power because of the small sample size available for the estimation of SNP-DNA methylation associations. Larger meQTL studies such as the GoDMC could not store association results for all SNPs (i.e., above a certain association *P*-value cutoff) because of technical reasons, and were therefore not usable for the two-sample reverse MR. Thus, the non-significant findings must be interpreted with care. In addition, an interpretation of the causal effect size is difficult, since the underlying genetic associations are calculated on the standard deviation scale of DNA methylation levels. Another potential limitation of the study is that eGFR, CKD, and UACR are phenotypes estimated from different underlying parameters and have multifactorial influences. Thus, we performed several analyses to ensure that our EWAS results were not driven by known confounders, including type 2 diabetes, a potential confounder of the relation between DNAm and kidney function. First, the EWAS associations in each cohort were adjusted for confounders to remove their effects within a cohort. Second, the EWAS were performed in each cohort separately and then meta-analyzed, which corresponds to an adjustment e.g., for the prevalence of diabetes across cohorts. Finally, we checked for associations of our replicated CpGs within published diabetes EWAS studies. Of all our replicated CpG associations, only the UACR-associated CpG cg18181703 at *SOCS3* showed an association with type 2 diabetes. This CpG was also associated with smoking status, BMI, and the blood levels of the soluble tumor necrosis factor receptor 2. Taking into account that our EWAS was also adjusted for smoking status and BMI, we assume the effects of cg18181703 on UACR to be at least partially independent from type 2 diabetes, BMI and smoking state. While we controlled for several of these known factors, other factors such as unmeasured covariates could not be explicitly adjusted for in the analyses and may influence the findings.

Further EWAS studies with increased sample sizes and with DNA methylation quantified from additional tissues as well as functional analyses are needed to extend our knowledge on regulatory mechanisms of kidney function, and to ultimately improve prediction and treatment of kidney disease. This holds true specifically for UACR, given the lower number of observed significant CpG associations.

In summary, this large-scale EWAS meta-analysis substantially extended the number of CpGs reproducibly associated with eGFR and CKD and revealed seven associations for UACR and microalbuminuria. DNA methylation at these sites explained a large proportion of eGFR variance, and differential methylation at four CpGs showed evidence for a potentially causal relationship to eGFR. Comprehensive characterization of replicated CpGs among patients with CKD, in kidney tissue, for differential gene expression, and for enriched pathways and epigenetic marks provide insights into kidney function associated transcriptional regulation.

## Methods

**Overview**. We set up a collaborative meta-analysis based on a distributive data model and quality-control procedures. To maximize phenotype standardization across studies, an analysis plan and a command line script (https://github.com/genepi-freiburg/ckdgen-pheno/tree/ckdgen-ewas-pheno) were created and provided to all participating studies (predominantly population-based studies; Supplementary Data 1 and 2). Automatically generated summary files were checked centrally. Upon phenotype approval, studies ran their EWAS and uploaded results and aggregated DNA methylation information to a central server. EWAS quality control was performed with custom scripts to assess inflation, positive controls, distribution of CpG probes and compare across studies the overall distributions of effect sizes, standard errors, and *P*-values. All study protocols were approved by the respective local ethics committees. All participants in all studies provided written informed consent.

**Phenotype definition**. Creatinine values obtained with Jaffé assay before 2009 were calibrated by multiplying by 0.95[52]. Studies estimated GFR with the Chronic Kidney Disease Epidemiology Collaboration (CKD-EPI) equation[53]. eGFR was winsorized at 15 and 200 ml min$^{-1}$ per 1.73 m$^2$. CKD was defined as an eGFR below 60 ml min$^{-1}$ per 1.73 m$^2$. The UACR values measured in mg/g were natural log transformed prior to all analyses. Microalbuminuria was defined as 1 for UACR > 30 mg/g and as 0 for UACR values < 10 mg/g.

**DNA methylation quantification and quality control**. For the quantification of DNA methylation genomic DNA was extracted from peripheral blood. Levels of DNA methylation were quantified using the Infinium MethylationEPIC BeadChip array (EPIC), the Illumina Infinium HumanMethylation450K BeadChip array (HM450K) or the Illumina Infinium HumanMethylation27 BeadChip array (HM27K). DNA methylation data preprocessing was performed according to individual study protocols including background correction, quantile normalization, probe filtering, sample filtering, SNP matching to the SNP control probe locations, outlier filtering and assay type correction (Supplementary Data 3). The methylation level at each site was represented and analyzed as β-value. CpG probes overlapping with SNPs were annotated. Each study computed mean and standard deviation of each CpG site and these summary statistics were compared between studies for systematic differences across CpGs and followed up with individual study analysts.

**Covariate assessment**. DNA methylation and covariates were measured at the same visit/time point. Prevalent diabetes was defined as fasting plasma glucose ≥ 126 mg/dl, non-fasting plasma glucose ≥ 200 mg/dl, treatment for diabetes, or self-report of a diabetes diagnosis. Prevalent hypertension was defined as systolic blood pressure ≥ 140 mm Hg, diastolic blood pressure ≥ 90 mm Hg, or treatment for hypertension. If measured blood pressure was not available, hypertension was defined by self-reporting. Current smoking status was defined using self-reported information. BMI (kg/m$^2$) was calculated using weight and height measurements as assessed in each study. Age was included as continuous values in the association models. Population structure in non-family studies was adjusted by genetic principal components (PC). White blood cell type proportions were estimated based on DNA methylation[54]. Additional technical covariates included control probe PCs[55], study center, processing batch, array sentrix ID, and sentrix position.

**Statistical methods and meta-analysis**. To ensure comparable power among analyzed sites only autosomal CpGs measured by both, EPIC and HM450K, were included in analyses. Each study performed linear regression analyses separated by ancestry groups. For assessing the robustness of the EWAS results, the analyses were limited to studies and subsamples of European ancestry individuals. DNA methylation β-values were modeled as the dependent variables with trait being either continuous eGFR or UACR values or binary CKD or microalbuminuria variables:

DNA methylation ~ trait + sex + age + genetic PCs + white blood cell proportions + technical covariates + diabetes + hypertension + BMI + current smoking

Participants needed complete information for all variables and study summary statistics and were included only if a minimum of 50 participants for eGFR/UACR and 50 cases/controls for CKD/microalbuminuria were available, respectively.

Each study-specific EWAS was adjusted for inflation prior to the meta-analysis by the BACON approach if the inflation estimate was greater or equal to one (Supplementary Fig. 8)[56]. Studies were split into discovery and replication by chronological order of contribution to the CKDGen Consortium meta-analysis (Supplementary Data 1 and 2). A fixed effect inverse-variance weighted meta-analysis as implemented in the R package 'metafor' (version 2.1-0) was performed for discovery studies, replication studies and for the resulting effect estimates of discovery and replication. CpGs were excluded if less than half the respective sample size was available within either discovery or replication or if the $I^2$ heterogeneity estimate was greater 95%. Successful replication of an associated CpG was defined as consistent direction of the effect estimates between discovery ($n_{\text{eGFR disc}} = 22,347$, $n_{\text{UACR disc}} = 11,458$) and replication ($n_{\text{eGFR repl}} = 11,258$, $n_{\text{UACR repl}} = 3610$) meta-analysis, a Bonferroni adjusted significance of the discovery P-value ($p_{\text{disc}}$) <1.1E−7 (#CpGs eGFR = 441,870, #CpGs UACR = 441,854), nominal significance of the replication P-value ($p_{\text{repl}}$) <0.05, and a combined discovery and replication P-value ($p_{\text{comb}}$) <1.1E−7.

**Gene expression analyses.** The effects of the kidney function trait-associated CpGs were tested for associations with gene expression in blood using two datasets: (1) mRNA levels of monocytes from 1202 participants of the MESA study, and (2) whole blood mRNA of 713 individuals of the KORA F4 study [57,58].

As initial step, a lookup of the CpG methylation levels with mRNA levels available in the association results from the MESA study was performed. For this lookup, association results with a P-value < 1E−5 were available. The analysis was described in detail in Kennedy et al.[59]. Briefly, gene expression was assessed using the Illumina HumanHT-12 v3.0 and v4.0 Expression BeadChips, and DNA methylation by the Illumina HM450K array. Expression values were normalized using the variance stabilizing transformation. 13,933 transcripts from the v3.0 and v4.0 arrays, which were significantly expressed above background levels (detection P-value < 0.01) in at least 5% of subjects. Association analyses in MESA were performed as a linear mixed model using log-transformed gene expression values as dependent variable, DNA methylation beta values as independent variable with age, sex, ethnicity, and study center added as covariates into the model.

A second association test was performed in the KORA F4 dataset. The association of the methylation level at replicated CpGs with gene expression levels of genes within ±500 kb vicinity was calculated using the log2-transformed mRNA levels obtained from the Illumina Human HT-12v3 gene expression array. The gene expression values were regressed on the DNA methylation beta values adjusted for sex and age. Prior to the analysis the technical factors as well as the blood cell type proportions were regressed out of the mRNA and DNA methylation levels, and its residuals were included in the final association model.

Annotation and quality control checks of the gene expression probes were based on the table provided in Schurmann et al.[60]. CpG-gene expression associations in blood that were available in the MESA results and had an association P-value<0.05 in KORA F4 with consistent effect direction were considered as significant. All gene expression probes passed the annotation-based quality control check.

**DNA methylation in kidney tissue.** The analyses using DNA methylation in kidney tissue with eGFR and fibrosis were performed using data from 506 microdissected kidney tissue samples using the Illumina EPIC BeadChip. The kidney tissue samples were collected separately, and are distinct from the blood samples that were analyzed in the EWAS meta-analysis. These samples were collected from unaffected portions of tumor nephrectomies and prepared as described before [61].

In brief, SeSAMe software[62] was used to perform preprocessing and quality control including low intensity-based detection, bleed-through correction in background subtraction, nonlinear dye bias correction, control for bisulfite conversion, calculation of beta values, and estimation of leukocyte fraction. Beta values of CpGs associated with eGFR and UACR and clinical information were extracted for association analysis.

A regression model was applied to test the associations of DNA methylation betas of the final CpGs as dependent variables with eGFR and fibrosis levels, respectively, as independent variables adjusted for sex, age, genetic PCs (1–5), diabetes status, hypertension status, BMI, array sentrix ID, sentrix position, and bisulfite conversion control and estimated leukocyte fraction.

**Targeted investigations of eGFR probes in CKD patients.** The association of the 69 eGFR-associated and validated CpGs from the general population with eGFR in the German Chronic Kidney Disease (GCKD) study was evaluated after correcting for the number of evaluated sites, and statistical significance was defined as P-value < 7.2E−4 (0.05/69). The GCKD study is a prospective observational study of patients with CKD[63]. Briefly, 5217 adult patients under nephrological care provided written informed consent and were enrolled from 2010 to 2012. Inclusion

criteria were eGFR between 30 and 60 ml min$^{-1}$ per 1.73 m$^2$ or an eGFR of >60 ml min$^{-1}$ per 1.73 m$^2$ with UACR > 300 mg g$^{-1}$ (or a urinary protein–creatinine ratio of >500 mg g$^{-1}$). Follow-up of the patients for clinical endpoints is still ongoing. Study endpoints are continuously recorded in a standardized fashion based on hospital discharge letters and death certificates, and include kidney-related events as well as death. Study design and the recruited study population are described in more detail in previous publications[63,64]. The GCKD Study was approved by local ethic committees and registered in the national registry for clinical studies (DRKS 00003971). A subset of 559 patients with CKD attributed to systemic lupus erythematosus, membranous nephropathy, focal-segmental glomerulosclerosis or autosomal-dominant polycystic kidney disease was selected for DNA methylation quantification and measured using the Infinium MethylationEPIC BeadChip array (EPIC). Association with eGFR was evaluated analogously to the main analysis (see Statistical methods and meta-analysis), apart from adjustment for smoking which was coded 0/1/2 for never-/ex-/current-smoker. To evaluate the association of DNA methylation with time to kidney failure from study entry, Cox regression models were fitted for each CpG, and analogously for a combined endpoint of kidney failure and acute kidney injury. Besides the DNA methylation predictor, models were adjusted for age, sex, and CKD subtype. The Cox regression model provides estimates for cause-specific hazard ratio (HR) in the presence of the competing events, i.e., any other death except kidney-related death. Subdistribution hazard analyses were additionally carried out in order to evaluate potential indirect effects via the competing event. The proportional hazards assumption was assessed based on scaled Schoenfeld residuals. Graphical assessment for the two associated CpGs revealed no evidence of major violations (Supplementary Fig. 9).

**Regional association plots and annotation.** The plots for Supplementary Fig. 5 were created using the 'Gviz'[65] and 'rtracklayer'[66] R packages. A maximum of 40 sites within 50,000 bp upstream or downstream of the CpG site of interest were included in the plot. If the interval contained more than the 40 sites, the plotted region was reduced to the distance of the furthest site plus 10,000 bp. The RefSeq Genes section is based on the UCSC NCBI RefSeq track with gene symbols from the 'org.Hs.eg.db' R package, the CpG Islands section is based on the UCSC CpG Islands track, the Roadmap chromHMM section was based on the Roadmaps 15-state chromHMM model of the fetal kidney epigenome (Roadmap Epigenome ID: E086)[67] and the Common SNPs section is based on the UCSC Common SNPs(151) track[68]. Finally, the CpG correlation plot at the bottom of the figure is based on data of the KORA F4 study DNA methylation samples using the Illumina HumanMethylation450 BeadChip array, with missing sites being colored light gray.

**Variance explained by DNA methylation.** The percentage of phenotypic variance explained by the 69 replicated CpGs associated with eGFR was estimated using data of 1,888 participants from the KORA FF4 study, the seven-year follow-up of the KORA F4 study[57,58]. The KORA FF4 was not part of the EWAS meta-analysis. However, 988 individuals included in the analysis of explained variance overlapped with the KORA F4 participants of the EWAS. In this dataset, the variance explained by all CpGs independently of the covariates was estimated as the difference in the $R^2$ of the base model including the CpGs and the one without. The base model was defined as kidney trait ~sex + age + white blood cell proportions + diabetes + hypertension + BMI + current smoking with kidney trait representing eGFR and UACR. For two eGFR-associated CpGs (cg06008406, cg20004659), no data was available in the KORA FF4 dataset.

**Bi-directional Mendelian randomization analysis.** In forward MR, using the 'TwoSampleMR' R package[69], we investigated potential causal effects of DNA methylation at the replicated CpGs on eGFR and UACR. MR utilizes genetic instruments to minimize bias due to confounding and reverse causation[70]. Genetic instruments for DNA methylation (meQTL) were available for 47 and five CpGs for eGFR and UACR, respectively, as previously identified by GoDMC in up to 27,750 individuals[42]. European ancestry summary GWAS data on eGFR[10] and UACR[13] were used as the respective outcome data. Filters were applied for meQTL inclusion ($p_{\text{SNP}} < 1E−5$ with DNA methylation in a ± 500kB *cis* region, linkage disequilibrium $R^2 < 0.2$ within a 1MB region, Steiger filtering, MAF > 0.05). We performed inverse-variance weighted MR as well as MR sensitivity analyses (simple mode, weighted mode, weighted median, and MR Egger), or triangulation by estimating the Wald ratio in case only a single instrument per CpG site was available[71–73]. The effect estimates obtained from MR depend on the units of the underlying datasets, and in this case correspond to a per unit change in one standard deviation of methylation levels on natural log-transformed eGFR, and standard deviation of natural log-transformed UACR, respectively.

In reverse MR, we examined potential causal effects of kidney function traits on DNA methylation, using the genome-wide significant SNPs from trans-ethnic GWAS on eGFR[10] and UACR[13] as genetic instruments for eGFR and UACR, respectively. To maximize power on outcome data, we performed a z-score meta-analysis of SNP-CpG associations from the KORA F4 ($n = 1662$) and FHS ($n = 3868$) studies. The combined effect estimates and their standard errors of the meQTL included in the MR were estimated from sample size, allele frequency and z-score[74]. Filters were applied for SNP inclusion (P-value < 5E−8 with kidney trait,

one-sided $P$-value < 0.05 with blood urea nitrogen for eGFR instruments, ancestry heterogeneity $P$-value ≥ 0.01, Steiger filtering, MAF > 0.05). We performed inverse-variance weighted MR with multiplicative random effects (because a sufficiently high number of instruments from different loci was available per trait)[51] and MR sensitivity analyses (simple mode, weighted mode, weighted median, and MR Egger)[71–73]. As an additional sensitivity analysis addressing pleiotropic variants, we excluded in total 35 instruments that were associated with type 2 diabetes mellitus in a recent GWAS[75] including 898,130 individuals: eleven SNPs that had an association $P$-value < 5E−8 with diabetes, and 24 additional instruments that were in linkage disequilibrium ($r^2$ > 0.2 within 1 Mb, 1000 G EUR reference panel) with such a diabetes-associated SNP. Linkage disequilibrium was assessed via LDlink[76]. For the reverse MR, the effect estimates provide the per-unit change in natural log-transformed eGFR, and standard deviation of natural log-transformed UACR, respectively, on standard deviation of DNA methylation levels.

$P$-value multiple testing adjustment according to Benjamini–Hochberg FDR < 0.05 was applied per kidney trait, and for forward and reverse MR separately[77]. Heterogeneity $P$-values were obtained from Q-statistics.

**Enrichment analyses.** To inform the potential functional effects of the associated CpGs, we assessed the enrichment of these CpGs in sites of DNase I or histone modification (H3K4me1, H3K4me3, H3K9me3, H3K27me3), gene sets based on GO terms and pathways in the KEGG and Reactome databases[32–35].

TFBS enrichment analyses were performed as previously described in detail[14]. Briefly, enrichment testing was assessed using eForge[78] using permutation with matching for gene- and CpG island-region localization when sampling. Data was sourced from either the ENCODE (125 samples) or Roadmap Epigenomics (299 samples) projects generated by the Hotspot method[67,79,80]. The CpGs that were associated with eGFR and UACR, respectively, at $P$-value < 1E−05 in the meta-analysis were used as input (Supplementary Data 6 and 7), and 10,000 resampling runs, an active proximity filter and considered FDR < 0.05 as significant (Supplementary Data 13 and 14). Histone mark enrichment analyses were performed analogously (Supplementary Data 15 and 16).

Enrichment in gene sets or pathways was assessed using the methylGSA package and R version 3.6.1[81]. The enrichment test method was methylglm implementing a logistic regression adjusting for the number of probes per gene and the autosomal background which overlaps 450k and EPIC arrays. Gene sets or pathways with 100–500 genes were tested (default setting). We considered a gene set or pathway to be significantly enriched at FDR < 0.05 correcting for multiple testing within each database using the Benjamini and Hochberg method (Supplementary Data 17 and 18)[77].

**Lookup for known EWAS associations with other traits.** The data of the EWAS Catalog (http://ewascatalog.org/) was downloaded on 12/09/2020 and used for subsequent analyses[82]. The replicated CpGs associated with the kidney function traits were looked-up in the EWAS Catalog dataset for known associations with published EWAS on other traits. From the EWAS Catalog, only sites having an association $P$-value below epigenome-wide significance as also applied in our study EWAS ($P$-value < 1.1E−7), and traits that included solely adults with a minimum sample size of $n$ = 1000 were considered. In the case of multiple EWAS for similar traits, only the one with the largest sample size was included in the lookup (Supplementary Data 19). Because the EWAS Catalog split several studies by discovery and replication results that differed from the replicated EWAS sites listed in the corresponding publications, we checked the published EWAS on kidney function and C-Reactive Protein (CRP) and thus excluded these traits from the catalog lookup. Considering that blood pressure is a risk factor for kidney function, we additionally performed a lookup in a recent large EWAS on blood pressure[83] which was not included in the EWAS Catalog at time of access.

**EPIC array only CpGs.** The up to 404,339 CpGs that were quantified by the EPIC array but not the 450k array were excluded from the analyses as the lower sample size combined with the sample split with approximately 2/3 replication samples would have distorted the findings and particularly the downstream analyses. To inform potential future meta-analyses with additional EPIC samples, the meta-analysis results of non-analyzed EPIC-specific CpG probes will be made available upon reasonable request, in addition to the publically available results (see Data availability).

**Reporting summary.** Further information on research design is available in the Nature Research Reporting Summary linked to this article.

## Data availability

The individual participant data included in this project are generally not publically available due to data protection laws, but can be applied from the individual studies on reasonable request. The summary statistics from the meta-analyses are available in the CKDGen Consortium website (https://ckdgen.imbi.uni-freiburg.de). For the lookup of the replicated CpGs in other EWAS, the publically available data of the EWAS Catalog (http://ewascatalog.org/, downloaded on 12/09/2020), the results of Richard et al. (PMID: 29198723), the results of Ligthart et al. (PMID: 27955697), and the results of Sheng et al. (PMID: 33144501) were included. Lookup of the CpGs with mRNA levels was conducted

using the published dataset of Kennedy et al. (PMID: 29914364), and for meQTLs the results provided by the GoDMC (http://www.godmc.org.uk).

## Code availability

The script for generating the phenotypes used in the EWAS is available via GitHub [https://github.com/genepi-freiburg/ckdgen-pheno-ewas][84]. EWAS QC, meta-analysis and postprocessing were implemented in R v4.0.1 using metafor v2.4.0, qqman v0.1.4, limma v3.42.2, openxlsx v4.1.5, car v3.0.8, bacon v1.16.0, mutoss v0.1.12, methylGSA v1.6.1, ggplot2 v3.3.3, SeSAMe v1.10.5 and rmeta v3.0.

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

## Acknowledgements
Acknowledgements and funding sources are listed in the Supplementary Text.

## Author contributions
Functional studies: Hongbo Liu, Katalin Susztak. Management of an individual contributing study: Adebowale A. Adeyemo, Nasir A. Aziz, Andrea Baccarelli, Murielle Bochud, Hermann Brenner, Monique M.B. Breteler, Layal Chaker, John C. Chambers, Shelley A. Cole, Josef Coresh, Tanguy Corre, Adolfo Correa, Simon R. Cox, Kai-Uwe Eckardt, Arif B. Ekici, Kathryn L. Evans, Christian Gieger, Megan L. Grove, Xīn Gào, Sarah E. Harris, Gibran Hemani, Peter Henneman, Lifang Hou, Mikko A. Hurme, Marjo-Riitta Jarvelin, Sharon L.R. Kardia, Marcus E. Kleber, Niek de Klein, Wolfgang Koenig, Jaspal S. Kooner, Holly Kramer, Florian Kronenberg, Anna Köttgen, Terho Lehtimäki, Daniel Levy, Lars Lind, Yongmei Liu, Donald M. Lloyd-Jones, Marie Loh, Riccardo E. Marioni, Karlijn A.C. Meeks, Joyce B.J. van Meurs, Lili Milani, Josine L. Min, Pashupati P. Mishra, Winfried März, Matthias Nauck, Ana Navas-Acien, Annette Peters, Bruce M. Psaty, Olli T. Raitakari, Alex P. Reiner, Sylvia E. Rosas, Joel Schwartz, Ben Schöttker, Jennifer A. Smith, Harold Snieder, Hannah R. Stocker, Johan Sundström, Maria Tellez-Plaza, Jana V. van Vliet-Ostaptchouk, Melanie Waldenberger, Juliane Winkelmann, Bruce H.R. Wolffenbuttel. Drafting of manuscript: Anna Köttgen, Pascal Schlosser, Alexander Teumer, Chris H.L. Thio, Adrienne Tin. Study design of an individual contributing study: Adebowale A. Adeyemo, Charles Agyemang, Andrea Baccarelli, Murielle Bochud, Hermann Brenner, Monique M.B. Breteler, Layal Chaker, John C. Chambers, Shelley A. Cole, Adolfo Correa, Simon R. Cox, Kai-Uwe Eckardt, Arif B. Ekici, Kathryn L. Evans, Myriam Fornage, Mohsen Ghanbari, Christian Gieger, Gibran Hemani, Peter Henneman, Marjo-Riitta Jarvelin, Sharon L.R. Kardia, Jaspal S. Kooner, Florian Kronenberg, Anna Köttgen, Daniel Levy, Ake T. Lu, Riccardo E. Marioni, Karlijn A.C. Meeks, Josine L. Min, Winfried März, Ana Navas-Acien, Annette Peters, Holger Prokisch, Bruce M. Psaty, Alex P. Reiner, Joel Schwartz, Jennifer A. Smith, Harold Snieder, Jana V. van Vliet-Ostaptchouk, Juliane Winkelmann, Bruce H.R. Wolffenbuttel, Johan Ärnlöv. Statistical methods and analysis: Nasir A. Aziz, Cristian Carmeli, John C. Chambers, Graciela E. Delgado, Arce Domingo, Lude Franke, Eliza Fraszczyk, Xu Gao, Sahar Ghasemi, Franziska Grundner-Culemann, Gibran Hemani, Peter Henneman, Anselm Hoppmann, Steve Horvath, Mikko A. Hurme, Shih-Jen Hwang, Roby Joehanes, Silva Kasela, Marcus E. Kleber, Anna Köttgen, Brigitte Kühnel, Terho Lehtimäki, Dan Liu, Hongbo Liu, Marie Loh, Kurt Lohman, Ake T. Lu, Pamela R. Matias-Garcia, Daniel L. McCartney, Karlijn A.C. Meeks, Josine L. Min, Pashupati P. Mishra, Christoph Nowak, Scott M. Ratliff, Sylvia E. Rosas, Pascal Schlosser, Sanaz Sedaghat, Brenton R. Swenson, Alexander Teumer, Chris H.L. Thio, Adrienne Tin, Andrea Venema, Niek Verweij, Rosie M. Walker, Antoine Weihs, Matthias Wielscher, Zhi Yu, Wei Zhao, Yinan Zheng. Subject recruitment: Charles Agyemang, Nasir A. Aziz, Andrea Baccarelli, Murielle Bochud, Hermann Brenner, John C. Chambers, Simon R. Cox, Lude Franke, Jaspal S. Kooner, Florian Kronenberg, Anna Köttgen, Lars Lind, Karlijn A.C. Meeks, Lili Milani, Matthias Nauck, Annette Peters, Joel Schwartz. Bioinformatics: Nasir A. Aziz, Xu Gao, Peter Henneman, Anselm Hoppmann, Roby Joehanes, Niek de Klein, Hongbo Liu, Ake T. Lu, Pamela R. Matias-Garcia, Daniel L. McCartney, Karlijn A.C. Meeks, Christoph Nowak, Pascal Schlosser, Andrea Venema, Antoine Weihs. Interpretation of results: Nasir A. Aziz, Andrea Baccarelli, Hermann Brenner, Layal Chaker, James S. Floyd, Xu Gao, Wolfgang Koenig, Anna Köttgen, Daniel Levy, Pamela R. Matias-Garcia, Ana Navas-Acien, Christoph Nowak, Bruce M. Psaty, Sylvia E. Rosas, Pascal Schlosser, Joel Schwartz, Ben Schöttker, Nona Sotoodehnia, Katalin Susztak, Maria Tellez-Plaza, Alexander Teumer, Chris H.L. Thio, Adrienne Tin, Niek Verweij. Methylation assessment: Charles Agyemang, Andrea Baccarelli, Hermann Brenner, Shelley A. Cole, James S. Floyd, Myriam Fornage, Xu Gao, Megan L. Grove, Xīn Gào, Sarah E. Harris, Peter Henneman, Steve Horvath, Lifang Hou, Mikko A. Hurme, Silva Kasela, Marcus E. Kleber, Stefan Lorkowski, Daniel L. McCartney, Joyce B.J. van Meurs, Lili Milani, Jennifer A. Smith, Nona Sotoodehnia, Silvia Stringhini, Alexander Teumer, Adrienne Tin, Jana V. van Vliet-Ostaptchouk, Melanie Waldenberger, Juliane Winkelmann, Bruce H.R. Wolffenbuttel, Wei Zhao, Yinan Zheng. Phenotyping: Charles Agyemang, Murielle Bochud, Hermann Brenner, Monique M.B. Breteler, Simon R. Cox, Kai-Uwe Eckardt, James S. Floyd, Sarah E. Harris, Christian Herder, Sharon L.R. Kardia, Silva Kasela, Wolfgang Koenig, Florian Kronenberg, Anna Köttgen, Joyce B.J. van Meurs, Annette Peters, Bruce M. Psaty, Scott M. Ratliff, Johan Ärnlöv. Critical review of manuscript: Adebowale A. Adeyemo, Charles Agyemang, Nasir A. Aziz, Andrea Baccarelli, Hermann Brenner, Monique M.B. Breteler, Cristian Carmeli, Layal Chaker, John C. Chambers, Josef Coresh, Adolfo Correa, Simon R. Cox, Graciela E. Delgado, Arif B. Ekici, Karlhans Endlich, Kathryn L. Evans, James S. Floyd, Myriam Fornage, Xu Gao, Mohsen Ghanbari, Christian Gieger, Philip Greenland, Megan L. Grove, Sarah E. Harris, Peter Henneman, Christian Herder, Lifang Hou, Silva Kasela, Marcus E. Kleber, Niek de Klein, Wolfgang Koenig, Jaspal S. Kooner, Florian Kronenberg, Anna Köttgen, Daniel Levy, Lars Lind, Marie Loh, Stefan Lorkowski, Riccardo E. Marioni, Pamela R. Matias-Garcia, Daniel L. McCartney, Karlijn A.C. Meeks, Lili Milani, Winfried März, Matthias Nauck, Ana Navas-Acien, Christoph Nowak, Holger Prokisch, Bruce M. Psaty, Alex P. Reiner, Sylvia E. Rosas, Pascal Schlosser, Joel Schwartz, Sanaz Sedaghat, Jennifer A. Smith, Nona Sotoodehnia, Hannah R. Stocker, Johan Sundström, Katalin Susztak, Maria Tellez-Plaza, Alexander Teumer, Chris H.L. Thio, Adrienne Tin, Niek Verweij, Melanie Waldenberger, Juliane Winkelmann, Yinan Zheng, Johan Ärnlöv.

## Funding

## Competing interests
Johan Ärnlöv has served on advisory boards for AstraZeneca and Boehringer Ingelheim, and have received lecturing fees from AstraZeneca and Novartis, all unrelated to the present project. Josef Coresh received grants from NIH and consultant to healthy.io. James S. Floyd has consulted from Shionogi Inc. Christian Herder reports personal fees from Sanofi and Lilly and grant support from Sanofi outside the submitted work. Marcus Kleber is employed with SYNLAB Holding Deutschland GmbH. Wolfgang Koenig reports personal fees from AstraZeneca, Novartis, Pfizer, The Medicines Company, DalCor, Kowa, Amgen, Corvidia, Daichii-Sankyo, Genentech, Novo Nordisk, Omeicos, Esperion, Berlin-Chemie, Sanofi, and Bristol-Myers Squibb and grants and non-financial support from Abbott, Roche Diagnostics, Beckmann, and Singulex outside the submitted work. S. Lorkowski reports grants and personal fees from Akcea Therapeutics Germany, and personal fees from amedes, AMGEN, Berlin-Chemie, Boehringer Ingelheim Pharma, Daiichi Sankyo, Lilly Deutschland, MSD Sharp & Dohme, Novo Nordisk Pharma, Roche Pharma, Sanofi-Aventis, Synlab Holding Deutschland, Unilever and Upfield, all outside the submitted work. Riccardo E. Marioni has received payment from Illumina for presentations. Winfried März reports grants from Siemens Healthineers, grants and personal fees from Aegerion Pharmaceuticals, grants and personal fees from AMGEN, grants from Astrazeneca, grants and personal fees from Sanofi, grants and personal fees from Alexion Pharmaceuticals, grants and personal fees from BASF, grants and personal fees from Abbott Diagnostics, grants and personal fees from Numares AG, grants and personal fees from Berlin-Chemie, grants and personal fees from Akzea Therapeutics, grants from Bayer Vital GmbH, grants from bestbion dx GmbH, grants from Boehringer Ingelheim Pharma GmbH Co KG, grants from Immundiagnostik GmbH, grants from Merck Chemicals GmbH, grants from MSD Sharp and Dohme GmbH, grants from Novartis Pharma GmbH, grants from Olink Proteomics, other from Synlab Holding Deutschland GmbH, all outside the submitted work. Bruce M. Psaty serves on the Steering Committee of the Yale Open Data Access Project funded by Johnson & Johnson. Sylvia Rosas receives research funding through her institution for clinical trials from Bayer and Astra Zeneca. She has participated in advisory boards for Reata and Relypsa. Johan Sundström reports ownership in companies providing services to Itrim, Amgen, Janssen, Novo Nordisk, Eli Lilly, Boehringer, Bayer, Pfizer and AstraZeneca, outside the submitted work. Niek Verweij is currently employee of Regeneron Genetics Center. All other authors report no competing interests.

## Additional information

Pascal Schlosser [1,2,111✉], Adrienne Tin [2,3,111], Pamela R. Matias-Garcia [4,5,6], Chris H. L. Thio [7], Roby Joehanes [8,9], Hongbo Liu [10], Antoine Weihs [11], Zhi Yu [12,13,14], Anselm Hoppmann [1], Franziska Grundner-Culemann [1], Josine L. Min [15,16], Adebowale A. Adeyemo [17], Charles Agyemang [18], Johan Ärnlöv [19,20], Nasir A. Aziz [21,22], Andrea Baccarelli [23], Murielle Bochud [24], Hermann Brenner [25,26,27,28], Monique M. B. Breteler [21,29], Cristian Carmeli [24,30], Layal Chaker [31,32], John C. Chambers [33,34,35,36], Shelley A. Cole [37], Josef Coresh [2], Tanguy Corre [24], Adolfo Correa [3], Simon R. Cox [38], Niek de Klein [39], Graciela E. Delgado [40], Arce Domingo-Relloso [41,42,43], Kai-Uwe Eckardt [44,45], Arif B. Ekici [46], Karlhans Endlich [47,48], Kathryn L. Evans [49], James S. Floyd [50,51,52], Myriam Fornage [53,54], Lude Franke [39,55], Eliza Fraszczyk [7], Xu Gao [23,56], Xīn Gào [25], Mohsen Ghanbari [31], Sahar Ghasemi [11,48,57], Christian Gieger [4,5], Philip Greenland [58], Megan L. Grove [54], Sarah E. Harris [38], Gibran Hemani [15,16], Peter Henneman [59], Christian Herder [60,61,62], Steve Horvath [63,64], Lifang Hou [58], Mikko A. Hurme [65], Shih-Jen Hwang [8,66], Marjo-Riitta Jarvelin [67,68,69,70], Sharon L. R. Kardia [71], Silva Kasela [72], Marcus E. Kleber [40,73], Wolfgang Koenig [74,75,76], Jaspal S. Kooner [35,36,77], Holly Kramer [78,79], Florian Kronenberg [80], Brigitte Kühnel [4,5], Terho Lehtimäki [81,82,83], Lars Lind [84], Dan Liu [21], Yongmei Liu [85], Donald M. Lloyd-Jones [58], Kurt Lohman [85], Stefan Lorkowski [86,87], Ake T. Lu [63], Riccardo E. Marioni [49], Winfried März [40,87,88,89], Daniel L. McCartney [49], Karlijn A. C. Meeks [17,18], Lili Milani [72], Pashupati P. Mishra [81,82,83], Matthias Nauck [48,90], Ana Navas-Acien [42], Christoph Nowak [19], Annette Peters [5,91], Holger Prokisch [92,93], Bruce M. Psaty [50,51,52,94], Olli T. Raitakari [95,96,97], Scott M. Ratliff [71], Alex P. Reiner [51], Sylvia E. Rosas [98,99], Ben Schöttker [25,26], Joel Schwartz [100], Sanaz Sedaghat [101], Jennifer A. Smith [71,102], Nona Sotoodehnia [52], Hannah R. Stocker [25,26], Silvia Stringhini [24], Johan Sundström [84,103], Brenton R. Swenson [52,104], Maria Tellez-Plaza [41], Joyce B. J. van Meurs [32], Jana V. van Vliet-Ostaptchouk [105], Andrea Venema [59], Niek Verweij [106], Rosie M. Walker [49], Matthias Wielscher [67], Juliane Winkelmann [92,107,108,109], Bruce H. R. Wolffenbuttel [105], Wei Zhao [71], Yinan Zheng [58], Estonian Biobank Research Team*, Genetics of DNA Methylation Consortium*, Marie Loh [33,34], Harold Snieder [7], Daniel Levy [8,9], Melanie Waldenberger [4,5,75], Katalin Susztak [10], Anna Köttgen [1,2,112] & Alexander Teumer [48,57,110,112✉]

[1]Institute of Genetic Epidemiology, Faculty of Medicine and Medical Center - University of Freiburg, Freiburg, Germany. [2]Department of Epidemiology, Johns Hopkins Bloomberg School of Public Health, Baltimore, MD, USA. [3]Department of Medicine, University of Mississippi Medical Center, Jackson, MS 39216, USA. [4]Research Unit Molecular Epidemiology, Helmholtz Zentrum München, German Research Center for Environmental Health, D-85764 Neuherberg, Bavaria, Germany. [5]Institute of Epidemiology, Helmholtz Zentrum München, German Research Center for Environmental Health, D-85764 Neuherberg, Bavaria, Germany. [6]TUM School of Medicine, Technical University of Munich, Munich, Germany. [7]Department of Epidemiology, University of Groningen, University Medical Center Groningen, Groningen, the Netherlands. [8]Framingham Heart Study, Framingham, Massachusetts, USA. [9]Population Sciences Branch, National Heart, Lung, and Blood Institute, National Institutes of Health, Bethesda, MD, US. [10]Department of Medicine and Genetics, University of Pennsylvania Perelman School of Medicine, Philadelphia, PA 19104, USA. [11]Department of Psychiatry and Psychotherapy, University Medicine Greifswald, Greifswald, Germany. [12]Program in Medical and Population Genetics, Broad Institute, Cambridge, MA, USA. [13]Cardiovascular Research Center, Massachusetts General Hospital, Boston, MA, USA. [14]Center for Genomic Medicine, Massachusetts General Hospital, Boston, MA, USA. [15]MRC Integrative Epidemiology Unit, University of Bristol, Bristol, UK. [16]Population Health Sciences, Bristol Medical School, University of Bristol, Bristol, UK. [17]Center for Research on Genomics and Global Health, National Human Genome Research Institute, National Institutes of Health, Bethesda, MD, USA. [18]Department of Public and Occupational Health, Amsterdam Public Health Research Institute, Amsterdam University Medical Centers, University of Amsterdam, 1105 AZ Amsterdam, the Netherlands. [19]Department of Neurobiology, Care Sciences and Society (NVS), Family Medicine and Primary Care Unit, Karolinska Institutet, Huddinge, Sweden. [20]School of Health and Social Studies, Dalarna University, Falun, Sweden. [21]Population Health Sciences, German Centre for Neurodegenerative Diseases (DZNE), Bonn, Germany. [22]Department of Neurology, Faculty of Medicine, University of Bonn, Bonn, Germany. [23]Laboratory of Environmental Precision Health, Mailman School of Public Health, Columbia University, New York, NY, USA. [24]Center for Primary Care and Public Health (Unisanté), University of Lausanne, Lausanne, Switzerland. [25]German Cancer Research Center (DKFZ), Division of Clinical Epidemiology and Aging Research, Heidelberg, Germany. [26]Network Aging Research, Heidelberg University, Heidelberg, Germany. [27]Division of Preventive Oncology, German Cancer Research Center (DKFZ) and National Center for Tumor Diseases (NCT), Heidelberg, Germany. [28]German Cancer Consortium, German Cancer Research Center (DKFZ), Heidelberg, Germany. [29]Institute for Medical Biometry, Informatics and Epidemiology (IMBIE), Faculty of Medicine, University of Bonn, Bonn, Germany. [30]Population Health Laboratory, University of Fribourg, Fribourg, Switzerland. [31]Department of Epidemiology, Erasmus University Medical Center, Rotterdam, the Netherlands. [32]Department of Internal Medicine, Erasmus Medical Center, Rotterdam, the Netherlands. [33]Lee Kong Chian School of Medicine, Nanyang Technological University, Singapore, Singapore. [34]Department of Epidemiology and Biostatistics, Imperial College London, London, UK. [35]Department of Cardiology, Ealing Hospital, London North West Healthcare NHS Trust, Southall, UK. [36]Imperial College Healthcare NHS Trust, London, UK. [37]Texas Biomedical Research Institute, San Antonio, USA. [38]Lothian Birth Cohorts Group, Department of Psychology, The University of Edinburgh, 7 George Square, Edinburgh EH8 9JZ, UK. [39]Department of Genetics, University Medical Center Groningen, University of Groningen, Hanzeplein 1, Groningen, the Netherlands. [40]Vth Department of Medicine, Medical Faculty Mannheim, Heidelberg University, Mannheim, Germany. [41]Department of Chronic Diseases Epidemiology,

National Center for Epidemiology, Carlos III Health Institute, Madrid, Spain. [42]Department of Environmental Health Sciences, Columbia University Mailman School of Public Health, New York, NY, USA. [43]Department of Statistics and Operations Research, University of Valencia, Valencia, Spain. [44]Department of Nephrology and Hypertension, University of Erlangen-Nürnberg, Erlangen, Germany. [45]Department of Nephrology and Medical Intensive Care, Charité - Universitätsmedizin Berlin, Berlin, Germany. [46]Institute of Human Genetics, Friedrich-Alexander-UniversitätErlangen-Nürnberg, 91054 Erlangen, Germany. [47]Department of Anatomy and Cell Biology, University Medicine Greifswald, Greifswald, Germany. [48]DZHK (German Centre for Cardiovascular Research), Partner Site Greifswald, Greifswald, Germany. [49]Centre for Genomic and Experimental Medicine, Institute of Genetics and Cancer, University of Edinburgh, Edinburgh EH4 2XU, UK. [50]Department of Medicine, University of Washington, Seattle, WA 98101, USA. [51]Department of Epidemiology, University of Washington, Seattle, WA 98101, USA. [52]Cardiovascular Health Research Unit, University of Washington, Seattle, WA 98101, USA. [53]Brown Foundation Institute of Molecular Medicine, McGovern Medical School, Houston, TX 77030, USA. [54]Human Genetics Center, Department of Epidemiology, Human Genetics and Environmental Sciences, School of Public Health, The University of Texas Health Science Center at Houston, Houston, TX 77030, USA. [55]Oncode Institute, Utrecht, the Netherlands. [56]Department of Occupational and Environmental Health Sciences, School of Public Health, Peking University, Beijing, China. [57]Institute for Community Medicine, University Medicine Greifswald, Greifswald, Germany. [58]Department of Preventive Medicine, Northwestern University Feinberg School of Medicine, Chicago, IL, USA. [59]Department of Clinical Genetics, Amsterdam Reproduction & Development Research Institute, Amsterdam University Medical Centres, University of Amsterdam, Amsterdam, the Netherlands. [60]Institute for Clinical Diabetology, German Diabetes Center, Leibniz Center for Diabetes Research at Heinrich Heine University Düsseldorf, Düsseldorf, Germany. [61]German Center for Diabetes Research, Munich-Neuherberg, Germany. [62]Department of Endocrinology and Diabetology, Medical Faculty and University Hospital Düsseldorf, Heinrich Heine University Düsseldorf, Düsseldorf, Germany. [63]Department of Human Genetics, David Geffen School of Medicine, University of California Los Angeles, Los Angeles, CA 90095, USA. [64]Biostatistics, Fielding School of Public Health, UCLA, Los Angeles, CA, USA. [65]Department of Microbiology and Immunology, Faculty of Medicine and Health Technology, Tampere University, Tampere 33014, Finland. [66]Division of Intramural Research, Population Sciences Branch, National Heart, Lung, and Blood Institute, National Institutes of Health, Bethesda, MD, USA. [67]Department of Epidemiology and Biostatistics, MRC–PHE Centre for Environment & Health, School of Public Health, Imperial College London, London, UK. [68]Center for Life Course Health Research, Faculty of Medicine, University of Oulu, 90014 Oulu, Finland. [69]Biocenter Oulu, University of Oulu, Oulu, Finland. [70]Unit of Primary Care, Oulu University Hospital, Oulu, Finland. [71]Department of Epidemiology, School of Public Health, University of Michigan, Ann Arbor, MI 48104, USA. [72]Estonian Genome Centre, Institute of Genomics, University of Tartu, Tartu, Estonia. [73]SYNLAB MVZ Humangenetik Mannheim, Mannheim, Germany. [74]Deutsches Herzzentrum München, Technische Universität München, Munich, Germany. [75]DZHK (German Centre for Cardiovascular Research), Partner site Munich Heart Alliance, Munich, Germany. [76]Institute of Epidemiology and Medical Biometry, University of Ulm, Ulm, Germany. [77]National Heart and Lung Institute, Imperial College London, London, UK. [78]Departments of Public Health Science and Medicine, Loyola University Chicago, Maywood, IL, USA. [79]Edward Hines VA Medical Center, Hines, IL, USA. [80]Institute of Genetic Epidemiology, Medical University of Innsbruck, Innsbruck, Austria. [81]Department of Clinical Chemistry, Faculty of Medicine and Health Technology, Tampere University, Tampere, Finland. [82]Finnish Cardiovascular Research Centre, Faculty of Medicine and Health Technology, Tampere University, Tampere, Finland. [83]Department of Clinical Chemistry, Fimlab Laboratories, Tampere, Finland. [84]Department of Medical Sciences, Uppsala University, Uppsala, Sweden. [85]Department of Medicine, Division of Cardiology, Duke Molecular Physiology Institute, Duke University School of Medicine, Durham, NC, USA. [86]Institute of Nutritional Sciences, Friedrich Schiller University Jena, Jena, Germany. [87]Competence Cluster for Nutrition and Cardiovascular Health (nutriCARD) Halle-Jena-Leipzig, Jena, Germany. [88]Synlab Academy, SYNLAB Holding Deutschland GmbH, Mannheim and Augsburg, Augsburg, Germany. [89]Clinical Institute of Medical and Chemical Laboratory Diagnostics, Medical University of Graz, Graz, Austria. [90]Institute of Clinical Chemistry and Laboratory Medicine, University Medicine Greifswald, Greifswald, Germany. [91]Ludwig-Maximilians Universität München, Munich, Germany. [92]Institute of Human Genetics, Klinikum rechts der Isar, Technische Universität München, Munich, Germany. [93]Department of Computational Health, Institute of Neurogenomics, Helmholtz Zentrum München, Munich, Germany. [94]Department of Health Services, University of Washington, Seattle, WA 98101, USA. [95]Research centre of Applied and Preventive Cardiovascular Medicine, University of Turku, Turku, Finland. [96]Department of Clinical Physiology and Nuclear Medicine, Turku University Hospital, Turku, Finland. [97]Centre for Population Health Research, University of Turku and Turku University Hospital, Turku, Finland. [98]Joslin Diabetes Center, Boston, MA, USA. [99]Harvard Medical School, Boston, MA, USA. [100]Department of Environmental Health, Harvard T.H. Chan School of Public Health, Boston, MA, USA. [101]Division of Epidemiology and Community Health, School of Public Health, University of Minnesota, Minneapolis, USA. [102]Survey Research Center, Institute for Social Research, University of Michigan, Ann Arbor, MI, USA. [103]The George Institute for Global Health, University of New South Wales, Sydney, Australia. [104]Institute for Public Health Genetics, University of Washington, Seattle, WA, USA. [105]Department of Endocrinology, University of Groningen, University Medical Center Groningen, Groningen, the Netherlands. [106]Department of Cardiology, University Medical Center Groningen, University of Groningen, Hanzeplein 1, Groningen, the Netherlands. [107]Institute of Neurogenomics, Helmholtz Zentrum München, Munich, Germany. [108]Chair Neurogenetics, Klinikum rechts der Isar, Technische Universität München, Munich, Germany. [109]Munich Cluster for Systems Neurology (SyNergy), Munich, Germany. [110]Department of Population Medicine and Lifestyle Diseases Prevention, Medical University of Bialystok, Bialystok, Poland. [111]These authors contributed equally: Pascal Schlosser, Adrienne Tin. [112]These authors jointly supervised this work: Anna Köttgen, Alexander Teumer. *Lists of authors and their affiliations appear at the end of the paper. ✉email: pascal.schlosser@uniklinik-freiburg.de; ateumer@uni-greifswald.de

## Estonian Biobank Research Team

Lili Milani[72]

A full list of members and their affiliations appears in the Supplementary Information.

## Genetics of DNA Methylation Consortium

Josine L. Min [15,16] & Gibran Hemani [15,16]

