## [Peer Review File · Nature Communications]

Epigenome-wide Association Studies Identify DNA Methylation Associated with Kidney Function and DamageReviewers' Comments:

Reviewer #1:

Remarks to the Author:

In their manuscript "Epigenome-wide association studies (EWAS) identify DNA methylation associated with kidney function and damage", Schlosser et al. studied associations between DNA methylation and kidney function-related traits.

They performed an EWAS for two measures of kidney function: estimated glomerular filtration rate (eGFR) in 33605 samples and urinary albumin-to-creatinine ratio (UACR) in 15068 samples. They identified 69 CpG sites associated with eGFR, out of which 60 novel and 7 novel with UACR.

They further studied if their findings are directionally consistent with clinical outcomes, tested if the findings replicate in kidney tissue, performed enrichment analyses, and evaluated causality using Mendelian randomization. Over all, the study is well conducted and reports robust associations between DNA methylation and kidney function measures and, potentially provides useful information for the field. However, the following issues should be addressed:

- It would be essential to have a table of characteristics of the pooled samples in the manuscript main body; the authors should indicate that discovery and replication samples are comparable in terms of the key characteristics.
- Regarding Mendelian randomization, pleiotropy and heterogeneity p-values need to be reported. Based on Supplementary Figure 7, for cg04864179, at least, there seems to be some heterogeneity, which warrants precaution when interpreting the causal findings. Also, it is not clearly defined what is deemed as a significant causal effect.
- Throughout Discussion, the authors give very little insights on the possible underlying biology explaining associations between DNA methylation and kidney function. I would anticipate a little speculation, at least, on their key finding: what could be a theoretical pathophysiological mechanism explaining the causal association between DNA methylation (in blood) near IRF5 and eGFR? This should include a short description of the known biological function of IRF5 and known phenotypic consequences of IRF5 variations and reflections of the prior literature with your current causal evidence (i.e., not just a short statement that there is a correlation between methylation and mRNA levels).
- It is not clear if the 506 microdissected kidney tissue samples are from one of the populations included in the EWAS or from a separate population – this needs to be clarified in Methods. If DNA methylation data in blood is available for individuals in the kidney tissue cohort, it would be helpful to determine correlations between DNA methylation in blood vs. kidney tissue to be able to better evaluate to what extent the EWAS results obtained in blood are generalizable to kidney tissue.

Other comments and/or minor corrections:

- The authors have selected an analytical approach to split the participating studies into discovery and replication by chronological order of contribution, which has its advantages. Compared with an approach to meta-analyze the results from all study populations and considering non-heterogeneous effect sizes as an internal replication, however, the splitting of the studies results in weaker statistical power for locus discovery. Not limiting sample size may be useful in particular for UACR, as the authors discuss that, compared to eGFR, larger sample sizes are required to reveal trait-associated CpGs.
- Figure 1: This could be partially due to the automated pdf conversion in the submission portal, but it is difficult to distinguish orange points from orange points with a black ring. It would be good to indicate novel sites more clearly (different color) from the previously reported ones. Crosses indicating replication with CKD/microalbuminuria are useful. It would also be helpful to include a legend giving an explanation for the different markings in the figure (as you have in Figure 4C and 4D for different colors).
- Figures 2 and 3: Labelling all gene names is useless in Figures 2A and 3A, as it makes the figure too busy/messy, and it is impossible to identify the corresponding points and gene labels. It would be

more useful to label only some of the key points that seem to deviate from the regression line.

Alternatively, a forest plot representation might be more informative, as it would allow readers to directly compare effect sizes and corresponding confidence intervals in EA vs. AA samples.

- Figure 4: Please adjust the font size so that it is readable in every figure. Currently it is impossible to read x-axis text due to small font size in 4C and 4D and, compared with 4A and 4B, the font size is extremely small.

Reviewer #2:

Remarks to the Author:

The manuscript entitled "Epigenome-wide Association Studies (EWAS) Identify DNA Methylation Associated with Kidney Function and Damage " by Schlosser et al. presents a comprehensive study exploring epigenetic association with Kidney Function and Damage and exploring the data from many different resources to get deeper insights. The manuscript is well written and structured. Methods are appropriate and relevant conclusions are drawn from the results. I have a few comments that the authors might need to address:

- (1) The individual study used in this meta-analysis seemed to be quite different in some characteristics, for example, the mean age, the prevalence of diabetes, hypertension, and CKD (Supplementary table 1), and these characteristics might have significant impact on the results of the meta-analysis. Authors need to consider how to address this issue in this study.
- (2) Are the discovery population and the replication population comparable in terms of some characteristics which might have the potential influence on the results and conclusions? How did the authors choose one specific study to be discovery population or replication population?
- (3) Line 315-322, the authors addressed "overlapped CpGs", it's better to identify them in the supplementary tables for readers to read through easily.
- (4) Line 353-356, the authors stated that "All seven CpGs associated with UACR were also significantly associated with microalbuminuria...", but Supplementary table 5 did not have this result.
- (5) Line 661-662, the authors used linear mixed model, how did the fix and random effect were defined in this model?
- (6) Number of decimal places should be consistent in one table, at least for one item. For example, In Supplementary table 1, Age, mean (SD) in LifeLinesDeep study is 45.9 (13.1) and in Lothian Birth Cohort 1921 (W1) study is 79.15 (0.57).
- (7) In supplementary table 1, Hypertension, % (n) in CHS (EA) study is 148 (55.4), however, n should not have decimal numbers.
- (8) In supplementary table 1, there were some figures like this: 27.1 [24.75, 30.02] (see BMI (kg/m²), mean (SD) in Rotterdam Study II), authors should notify what the figures in square brackets mean.
- (9) All the abbreviations should be given their full name in the table and figure legends, or follow the requirement of the figure and table instructions of the journal. For example, what did "MA" mean in Supplementary table 2?
- (10) In Supplementary tables, for the comparison results, sometimes, authors coded the results as TRUE/FALSE (for example the last column in Supplementary table 7), sometimes, coded as 0/1 (for example the last two column in Supplementary table 10). This should be consistent in tables or at least providing the meaning of 0/1.
- (11) There are a few instances of typo errors which should be corrected.

Reviewer #3:

Remarks to the Author:

This is a very well done paper examining DNA methylation and kidney disease in a large number of cohorts, with excellent validation and functional assessments, particularly the addition of DNAm

assessments in microdissected kidneys, even though they did not align well with the blood data. I have few comments on the methods or results as presented, but one major one about interpretation of causality.

There are many statistical causality tests presented which could use better rationale for the biological mechanism by which the authors propose that DNA methylation changes in blood can directionally influence kidney function. The associative tests and enrichment analysis are all well done and extremely well validated, but the causality as discussed could be better supported by a hypothetical mechanism, and the assumptions of the causality tests need to be interpreted from the point of view of that hypothesis. This is a tricky line to walk in DNAm studies, but an important one if the authors wish to make strong causal conclusions, as they have done. In particular, the analysis of gene expression in blood and using that to promote causal relationships with kidney disease could be better explored, as could the overall failure of DNAm changes in blood to replicate in kidney samples.

Related, it is not always clear what the effect sizes are, and they are often not discussed in the text. From figures, they are incredibly small, but because the units are not mentioned (are they per eGFR unit changes or changes over the range of eGFR, for example), it is not clear just how small they really are. Given the magnitude of the effects and the question above about the biological mechanism by which blood DNAm can influence kidney, there are other possible conclusions from this same data. One is that an independent exposure or influence is responsible for both the DNAm changes and they kidney disease, and the low number of sites that pass MR analysis do give this some credit. Another is that what is being detected as DNAm differences in blood is in fact cell free DNA from kidney cells, which is increased in individuals with more severe CKD as more kidney cells die. From the methods this appears to be whole blood, and cell free DNA would be present at low levels, which is consistent with the apparent small effect sizes.

A brief comment about the diabetes variable – diabetes is known to have very strong effects on DNAm and is prevalent in CKD populations as shown in ST1/ST2. A sensitivity analysis even in a subset of the studies would go a long way to encouraging readers that diabetes status is not confounding.

Overall, I commend the authors for a very well executed analysis plan, and believe that it is possible to connect their results with a plausible biological hypothesis that would strengthen this manuscript.

Other comments:

- What is the source of the kidney DNAm data? This is not described.
- For the supplementary table with cohort data, it would be helpful to add tissue type – whole blood vs PBMCs, if applicable, as it is not clear from the methods.
- Were effect directions the same across all cohorts? In Table 1, it would be helpful to show the direction in all cohorts, similar to how the PACE consortium reports multi-cohort effects (ie PMID: 31015461, supplement)
- The 33k figure seems a bit disingenuous, since it was two separate EWASes, one with 22k for one measure and 11k for the other, which were not ever really combined.
- Hypomethylation has a specific definition that typically means less than 10% methylation, “lower methylation” is usually the correct term to refer to differences between groups.
- The “trans ancestry” analysis is clearly driven by Europeans, so is it really correct to describe it as trans-ancestry?
- X axis in Figure 3 is very small
- Line 524 typo, momber should be member
- Line 540-541 typos, “both concordantial implicated” is grammatically incorrect and waist should be waste
- Paragraph at line 542 could use editing for grammar and punctuation

Dear Reviewers,

Thank you for the thorough evaluation of our manuscript, and for your helpful and constructive comments. We have carefully considered them and responded to each comment in the point-by-point response, below.

We believe that your comments have helped us to significantly improve our manuscript. We hope that the implemented changes meet your requests, and that we have answered all open questions.

Sincerely,

Pascal Schlosser, Adrienne Tin, Anna Köttgen, and Alexander Teumer, on behalf of all authors

Point-by-point response

Reviewer #1:

Remarks to the Author:

In their manuscript "Epigenome-wide association studies (EWAS) identify DNA methylation associated with kidney function and damage", Schlosser et al. studied associations between DNA methylation and kidney function-related traits.

They performed an EWAS for two measures of kidney function: estimated glomerular filtration rate (eGFR) in 33605 samples and urinary albumin-to-creatinine ratio (UACR) in 15068 samples. They identified 69 CpG sites associated with eGFR, out of which 60 novel and 7 novel with UACR.

They further studied if their findings are directionally consistent with clinical outcomes, tested if the findings replicate in kidney tissue, performed enrichment analyses, and evaluated causality using Mendelian randomization. Over all, the study is well conducted and reports robust associations between DNA methylation and kidney function measures and, potentially provides useful information for the field.

Response R1R1: We thank the Reviewer for recognizing the scope of our work and thank the Review for the positive feedback.

However, the following issues should be addressed:

- It would be essential to have a table of characteristics of the pooled samples in the manuscript main body; the authors should indicate that discovery and replication samples are comparable in terms of the key characteristics.

Response R1R2: We added pooled sample characteristics for the discovery and replication datasets for eGFR and UACR as **Table 1**.

- Regarding Mendelian randomization, pleiotropy and heterogeneity p-values need to be reported. Based on Supplementary Figure 7, for cg04864179, at least, there seems to be some heterogeneity, which warrants precaution when interpreting the causal findings. Also, it is not clearly defined what is deemed as a significant causal effect.

Response R1R3: Thank you for raising this important issue. We added the heterogeneity p-values to the **Supplementary Tables 11 and 12**. A causal effect estimate of the primary results that passed the multiple testing correction at $FDR < 0.05$ (calculated per forward and reverse MR separately) was deemed as significant. We clarified this in the Results, and now refer to the Methods, where this was described in detail.

Among the four significant MR results, (only) cg04864179 showed significant heterogeneity ($p < 0.05$). Although the effect estimates were direction-consistent and of the same order of magnitude in the sensitivity analyses (simple mode, weighted mode, weighted median, and MR Egger), we agree that this causal effect should be interpreted with caution. We now discuss this aspect in the revised manuscript (Discussion, p. 23).

- Throughout Discussion, the authors give very little insights on the possible underlying biology explaining associations between DNA methylation and kidney function. I would anticipate a little speculation, at least, on their key finding: what could be a theoretical pathophysiological mechanism explaining the causal association between DNA methylation (in blood) near *IRF5* and eGFR? This should include a short description of the known biological function of *IRF5* and known phenotypic consequences of *IRF5* variations and reflections of the prior literature with your current causal evidence (i.e., not just a short statement that there is a correlation between methylation and mRNA levels).

Response R1R4: *IRF5* encodes a member of the interferon regulatory factor family and plays a critical role in innate immunity by activating expression of type I interferon (IFN). Although there is no explicit description of a direct effect of *IRF5* DNA methylation on eGFR in the literature, we hypothesize that there might be a relation via the IFN pathway. This relation is supported by several publications that connect *IRF5* expression to lupus nephritis as a manifestation of lupus erythematosus, thereby connect *IRF5* regulation to kidney pathology. Moreover, inhibition of *IRF5* hyperactivation resulted in reduced lupus severity and improved kidney pathology in a mouse model, further supporting the link of *IRF5* with kidney disease and pathology. We added this possible interpretation to the Discussion (p. 23).

- It is not clear if the 506 microdissected kidney tissue samples are from one of the populations included in the EWAS or from a separate population – this needs to be clarified in Methods. If DNA methylation data in blood is available for individuals in the kidney tissue cohort, it would be helpful to determine correlations between DNA methylation in blood vs. kidney tissue to be able to better evaluate to what extent the EWAS results obtained in blood are generalizable to kidney tissue.

Response R1R5: We have collected an unaffected portion of tumor nephrectomies. These kidney tissue samples were collected separately from the blood samples that were analyzed

as part of the meta-EWAS. In addition, there were no overlapping datasets that include both kidney and blood tissues of the same samples. The methods of samples collection and DNA methylation analysis of the kidney tissue samples included in our analyses were performed following out previous publication of Gluck *et al.* (Nat Comm, 2019).¹ We extended the method section to cover the specific source of the kidney tissue samples.

Other comments and/or minor corrections:

- The authors have selected an analytical approach to split the participating studies into discovery and replication by chronological order of contribution, which has its advantages. Compared with an approach to meta-analyze the results from all study populations and considering non-heterogeneous effect sizes as an internal replication, however, the splitting of the studies results in weaker statistical power for locus discovery. Not limiting sample size may be useful in particular for UACR, as the authors discuss that, compared to eGFR, larger sample sizes are required to reveal trait-associated CpGs.

Response R1R6: We fully agree that a combined meta-analysis of including both the discovery and replication sample would achieve the highest statistical power to reveal associations. Therefore, (and in addition to providing the full summary statistics of this EWAS meta-analysis) we included all results with an association $p\text{-value} < 1E-5$ in the combined meta-analysis as **Supplementary Tables 6 and 7**. In addition, we now added a column indicating whether a CpG is a replicated site in the 2-stage approach. However, we preferred a more conservative and well established discovery+replication approach to extract and focus on statistically robust results for subsequent analyses in our project.

Although the suggestion for considering non-heterogeneous effect sizes as an internal replication is interesting, it addresses a different statistical aspect of the results, and does not concur very well with the replication status as seen in the updated **Supplementary Tables 6 and 7**.

In the revised manuscript, we emphasized that the **Supplementary Tables 6 and 7** include the combined meta-analysis (p. 12).

- Figure 1: This could be partially due to the automated pdf conversion in the submission portal, but it is difficult to distinguish orange points from orange points with a black ring. It would be good to indicate novel sites more clearly (different color) from the previously reported ones. Crosses indicating replication with CKD/microalbuminuria are useful. It would also be helpful to include a legend giving an explanation for the different markings in the figure (as you have in Figure 4C and 4D for different colors).

Response R1R7: Thank you. We improved visibility by color coding the novel CpGs and adding a legend as you suggested.

- Figures 2 and 3: Labelling all gene names is useless in Figures 2A and 3A, as it makes the figure too busy/messy, and it is impossible to identify the corresponding points and gene labels. It would be more useful to label only some of the key points that seem to deviate from the regression line.

Alternatively, a forest plot representation might be more informative, as it would allow readers to directly compare effect sizes and corresponding confidence intervals in EA vs. AA samples.

Response R1R8: Based on the Reviewer's helpful suggestion, we labeled in **Figure 3A** only sites that indicated a significant between-ancestry heterogeneity after adjusting for the number of eGFR-associated sites tested ($P\text{-value} < 0.05/69$). In addition to the regression line, we added the diagonal to better visualize the deviation of the AA specific effects from EA. We added the results of the heterogeneity test to the Results (p. 12).

Figure 3A now labels only sites that were not nominally significantly associated with CKD ($P\text{-value} \geq 0.05$) to indicate a potential null effect in CKD.

Given that a direct comparison of the ancestry-specific effect sizes is presented in **Supplementary Tables 4 and 5**, we refrained from including additional forest plots for all associations to the manuscript.

- Figure 4: Please adjust the font size so that it is readable in every figure. Currently it is impossible to read x-axis text due to small font size in 4C and 4D and, compared with 4A and 4B, the font size is extremely small.

Response R1R9: Thank you. We tilted the labels and increased the font size such that we can fit the page width to increase the readability.

Reviewer #2:

Remarks to the Author:

The manuscript entitled "Epigenome-wide Association Studies (EWAS) Identify DNA Methylation Associated with Kidney Function and Damage" by Schlosser et al. presents a comprehensive study exploring epigenetic association with Kidney Function and Damage and exploring the data from many different resources to get deeper insights. The manuscript is well written and structured. Methods are appropriate and relevant conclusions are drawn from the results.

Response R2R0: Many thanks for the positive feedback.

I have a few comments that the authors might need to address:

(1) The individual study used in this meta-analysis seemed to be quite different in some characteristics, for example, the mean age, the prevalence of diabetes, hypertension, and CKD (Supplementary table 1), and these characteristics might have significant impact on the results of the meta-analysis. Authors need to consider how to address this issue in this study.

Response R2R1: We agree that there are several important patient characteristics that may confound the association of eGFR/UACR and DNAm, such as age, BMI, prevalence of diabetes, hypertension, smoking, and sex. We therefore chose to include these as co-variables

in our regression models, in addition to white blood cell proportions and technical covariates, which represents a more comprehensive adjustment than what was performed in earlier studies, e.g. Chu *et al.*² In addition, we conducted a look-up of all replicated CpG sites in the EWAS catalog and in specific EWAS publications that include summary statistics for all of the above listed traits. One overlapping finding with a diabetes EWAS is also discussed in more detail in **R3R3**.

Furthermore, we now included a new table with the pooled patient characteristics of the discovery and replication cohorts (see also **R1R2**; **Table 1**). Moreover, we tested for differences in the patient characteristics in between discovery and replication (P-value<0.05/6). Due to the large sample sizes of our analyses, we observed significant difference for eGFR for sex, diabetes, hypertension and smoking and for UACR for sex, diabetes and smoking. For each significant combination, we conducted a univariate meta-regression with the estimated effect size of our EWAS as the outcome and the respective covariate modelled as a moderator on order to study the impact of study-specific trait prevalence on the effect estimate. When testing all replicated CpG sites for eGFR (P-value<0.05/69) and UACR (P-value<0.05/7), we observed no significant moderator effects. Hence, these results indicate that the initial adjustment and stratified analysis across cohorts and ancestries, i.e., analysis within each ancestry-specific sub-cohort followed by meta-analysis, was sufficient to address potential confounding effects. The few CpG sites with overlapping findings in EWAS for other traits like diabetes have potentially pleiotropic effects, which are at least partially independent from eGFR and UACR. We explained this now in detail in the Discussion (p. 26).

(2) Are the discovery population and the replication population comparable in terms of some characteristics which might have the potential influence on the results and conclusions? How did the authors choose one specific study to be discovery population or replication population?

Response R2R2: We primarily focused on population-based studies in order to reduce heterogeneity related to the distribution of outcomes and confounders. However, there is a tradeoff between homogeneity and larger sample size to increase statistical power for the EWAS. Therefore, we decided to include also some comparable although not completely population-based cohorts.

We assigned the cohorts to discovery and replication stage consecutively. In detail, we collected all available and comparable cohorts for a discovery stage at project outset. After the discovery stage was completed, we sought additional replication cohorts that in the meantime had collected appropriate data to reduce the number of false positive association results (please see also R1R6).

The datasets used for discovery and replication are generally comparable in their trait distributions (please see R1R2), although small differences may achieve statistical significance given the overall large sample sizes. It is therefore possible that studies with a specifically high or low prevalence of a potential confounder would receive a higher weight in a meta-analysis in the smaller replication dataset. Since our manuscript focusses on associations that were replicated despite such potential differences, our main findings can be regarded as very robust.

(3) Line 315-322, the authors addressed "overlapped CpGs", it's better to identify them in the supplementary tables for readers to read through easily.

Response R2R3: We amended the **Supplementary Tables 6 and 7** with columns indicating the overlapping CpGs (Overlap with suggestive eGFR CpGs; Overlap with suggestive UACR CpGs).

(4) Line 353-356, the authors stated that "All seven CpGs associated with UACR were also significantly associated with microalbuminuria...", but Supplementary table 5 did not have this result.

Response R2R4: In **Supplementary Table 5** we reported the associations with microalbuminuria in columns AK to AQ. As can be seen from the meta-analysis P-values (AM) and the newly added effect-direction-consistency-check (AQ), all seven CpGs replicated (P-value < 0.05/7 = 0.007). As **Supplementary Table 5** is already quite comprehensive, we opted to not include an additional column indicating the replication status (in addition to the added effect-direction-consistency-column).

(5) Line 661-662, the authors used linear mixed model, how did the fix and random effect were defined in this model?

Response R2R5: According to the details provided in Kennedy *et al.* referenced to in the Methods section, the inter-sample correlation emended (ICE) was applied to correct for technical batch effects related to the gene expression data. This method estimates a covariance matrix of the gene expression values across all mRNA arrays, which is subsequently included for the association analysis as a random effect in the generalized linear model.³ In this way, the association model is corrected for overall technical confounding factors affecting the measured mRNA levels. All other listed covariates of the CpG-mRNA association were included as fixed effects.

(6) Number of decimal places should be consistent in one table, at least for one item. For example, In Supplementary table 1, Age, mean (SD) in LifeLinesDeep study is 45.9 (13.1) and in Lothian Birth Cohort 1921 (W1) study is 79.15 (0.57).

(7) In supplementary table 1, Hypertension, % (n) in CHS (EA) study is 148 (55.4), however, n should not have decimal numbers.

(8) In supplementary table 1, there were some figures like this: 27.1 [24.75, 30.02] (see BMI (kg/m²), mean (SD) in Rotterdam Study II), authors should notify what the figures in square brackets mean.

Response R2R6: Thank you for your thorough review. We implemented consistent rounding for **Supplementary Table 1 and 2** and double checked all reported cohort characteristics with the individual cohorts.

(9) All the abbreviations should be given their full name in the table and figure legends, or follow the requirement of the figure and table instructions of the journal. For example, what did “MA” mean in Supplementary table 2?

Response R2R7: MA stands for microalbuminuria. We now explained this and all other abbreviations of the (Supplementary) Tables and Figures.

(10) In Supplementary tables, for the comparison results, sometimes, authors coded the results as TRUE/FALSE (for example the last column in Supplementary table 7), sometimes, coded as 0/1 (for example the last two column in Supplementary table 10). This should be consistent in tables or at least providing the meaning of 0/1.

Response R2R8: Thank you for noticing this inconsistency. We now use True/False as labels in all Supplementary Tables where applicable.

(11) There are a few instances of typo errors which should be corrected.

Response R2R9: After completing the revision, we proofread the manuscript and fixed several typos.

Reviewer #3:

Remarks to the Author:

This is a very well done paper examining DNA methylation and kidney disease in a large number of cohorts, with excellent validation and functional assessments, particularly the addition of DNAm assessments in microdissected kidneys, even though they did not align well with the blood data. I have few comments on the methods or results as presented, but one major one about interpretation of causality.

Response R3R1: We thank the Reviewer for the positive evaluation of our manuscript.

There are many statistical causality tests presented which could use better rationale for the biological mechanism by which the authors propose that DNA methylation changes in blood can directionally influence kidney function. The associative tests and enrichment analysis are all well done and extremely well validated, but the causality as discussed could be better supported by a hypothetical mechanism, and the assumptions of the causality tests need to be interpreted from the point of view of that hypothesis. This is a tricky line to walk in DNAm studies, but an important one if the authors wish to make strong causal conclusions, as they have done. In particular, the analysis of gene expression in blood and using that to promote causal relationships with kidney disease could be better explored, as could the overall failure of DNAm changes in blood to replicate in kidney samples.

Related, it is not always clear what the effect sizes are, and they are often not discussed in the text. From figures, they are incredibly small, but because the units are not mentioned (are they

per eGFR unit changes or changes over the range of eGFR, for example), it is not clear just how small they really are. Given the magnitude of the effects and the question above about the biological mechanism by which blood DNAm can influence kidney, there are other possible conclusions from this same data. One is that an independent exposure or influence is responsible for both the DNAm changes and they kidney disease, and the low number of sites that pass MR analysis do give this some credit. Another is that what is being detected as DNAm differences in blood is in fact cell free DNA from kidney cells, which is increased in individuals with more severe CKD as more kidney cells die. From the methods this appears to be whole blood, and cell free DNA would be present at low levels, which is consistent with the apparent small effect sizes.

Response R3R2: The causal conclusions are based on statistical inference with all stated limitations. However, we added the discussion of a plausible hypothetical mechanism that connects *IRF5* methylation and eGFR as an example, which combines statistical evidence with hypothesis-driven mechanistic insights from the literature (please see also R1R4). Based on published work, *IRF5* acts through gene expression in immune cells and immune pathways on kidney function.

The interpretation of the causal effect sizes depends on the units and trait transformations of the underlying genetic associations for both exposure and outcome. We added the missing information to the methods (p. 36 and 37). However, the interpretation of the causal effect estimates might be of secondary importance as they do not reflect a causal effect of a clinical intervention but rather the lifelong exposure of the genetically determined differences of meQTLs. Mendelian randomization guidelines suggest that estimating the causal effect of the exposure on the outcome is less important and may even be unnecessary.⁴ Nevertheless, we converted the effect estimates (which are of the same magnitude for all results in **Table 3**, i.e. the former **Table 2**) exemplarily for the *IRF5* example, and added it to the discussion (p. 23): an increase in DNA methylation of ten standard deviations resulted in 3.2% higher eGFR. Furthermore, we added to the limitations the difficulty of interpreting the causal effect estimates because the genetic associations are calculated on the standard deviation scale of DNA methylation levels (p. 25). We added the interpretation of the effect estimates to the description of **Table 3**.

Our non-significant MR results are in line with the interpretation of an exposure that independently influences DNA methylation and kidney disease.⁵ We believe that any cell-free DNA from kidney in blood is less likely to explain our findings, because our data is largely derived from population-based studies, where very few individuals suffer from advanced kidney damage. In addition, several of our cohorts measured DNAm from PBMC (FHS, LifeLinesDeep) or even monocytes (MESA) instead of whole blood, which has been added to **Supplementary Table 3** now. The association effects of the three CpG sites that are also associated with eGFR in kidney tissue are not attenuated in PBMCs and monocytes compared to the results from whole blood (**Reviewer Figure 1**).

Reviewer Figure 1: Forest plots of the EWAS results (combined meta-analysis) of the three CpG sites that are also associated with eGFR in kidney tissue.

A brief comment about the diabetes variable – diabetes is known to have very strong effects on DNAm and is prevalent in CKD populations as shown in ST1/ST2. A sensitivity analysis even in a subset of the studies would go a long way to encouraging readers that diabetes status is not confounding.

Response R3R3: We fully agree that diabetes represents a potential confounder of the relation between DNAm and kidney function. As sensitivity analyses in subgroups are often difficult to interpret because of reduced sample size and statistical power, we had already designed our primary study to account for the effects of potential confounders in several ways: First, the EWAS associations in each cohort were adjusted for confounders to remove their effects within a cohort. Second, the EWAS were performed in each cohort separately and then meta-analyzed, which corresponds to an adjustment e.g., for the prevalence of diabetes across cohorts. Finally, we checked for associations of our replicated CpGs within published diabetes EWAS studies. Of all our replicated CpG associations, only the UACR-associated CpG cg18181703 at *SOCS3* showed an association with type 2 diabetes. This CpG was also associated with smoking status, BMI, and the blood levels of the soluble tumor necrosis factor receptor 2. Taking into account that our EWAS was also adjusted for smoking status and BMI, we assume the effects of cg18181703 on UACR to be at least partially independent from type 2 diabetes, BMI and smoking state. We added this description for removing confounding effects of diabetes (and other potential confounders) to the Discussion (p. 26).

Overall, I commend the authors for a very well executed analysis plan, and believe that it is possible to connect their results with a plausible biological hypothesis that would strengthen this manuscript.

Other comments:

- What is the source of the kidney DNAm data? This is not described.

Response R3R4: We have collected an unaffected portion of tumor nephrectomies and extended the methods to cover the specific source of the kidney tissue samples.

- For the supplementary table with cohort data, it would be helpful to add tissue type – whole blood vs PBMCs, if applicable, as it is not clear from the methods.

Response R3R5: We now collected and added the specific tissue types for each cohort and added them to **Supplementary Table 3**.

- Were effect directions the same across all cohorts? In Table 1, it would be helpful to show the direction in all cohorts, similar to how the PACE consortium reports multi-cohort effects (ie PMID: 31015461, supplement)

Response R3R6: Taking into account that the (former) Table 1 in the main text is already very wide, we added the proposed column providing the direction of the individual studies effect estimates to **Supplementary Tables 4 and 5**. While adding this information for UACR, we realized that one study was erroneously listed in the study description table for UACR (**Supplementary Table 2**), but not included in any of the presented results. The study was removed from Supplementary Table 2 in the revision. The results of the EWAS including the provided sample size were not affected.

- The 33k figure seems a bit disingenuous, since it was two separate EWASes, one with 22k for one measure and 11k for the other, which were not ever really combined.

Response R3R7: The eGFR meta-analysis of discovery and replication combines up to 33,605 individuals. These do overlap with the 15,068 individuals that were meta-analyzed for UACR. We reviewed the section where we reported on the 22k eGFR discovery and 11k eGFR replication analyses and indicated more clearly which sample size belongs to which analysis.

• Hypomethylation has a specific definition that typically means less than 10% methylation, “lower methylation” is usually the correct term to refer to differences between groups.

Response R3R8: We re-phrased this accordingly in the revised manuscript.

• The “trans ancestry” analysis is clearly driven by Europeans, so is it really correct to describe it as trans-ancestry?

Response R3R9: We agree that the results might be driven by Europeans, although we included a substantial amount of non-European ancestry individuals in our study as listed in **Supplementary Tables 1 and 2**. However, we now addressed that the results might be driven by European ancestry as a limitation in the Discussion (p. 21).

• X axis in Figure 3 is very small

Response R3R10: Thank you for pointing this out. We realized a typo in the unit of x-axis of both **Figure 2 and 3**, which should be [%methylation] not [sd of %methylation]. The x-axis reflects now correctly the scale of the effect sizes obtained from the corresponding EWAS that are presented in **Supplementary Tables 4 and 5**.

• Line 524 typo, momber should be member

• Line 540-541 typos, “both concordantial implicated” is grammatically incorrect and waist should be waste

• Paragraph at line 542 could use editing for grammar and punctuation

Response R3R11: Many thanks, we corrected these typos.

References:

1. Gluck C, *et al.* Kidney cytosine methylation changes improve renal function decline estimation in patients with diabetic kidney disease. *Nature communications* **10**, 2461 (2019).
2. Chu AY, *et al.* Epigenome-wide association studies identify DNA methylation associated with kidney function. *Nature communications* **8**, 1286 (2017).
3. Kang HM, Ye C, Eskin E. Accurate discovery of expression quantitative trait loci under confounding from spurious and genuine regulatory hotspots. *Genetics* **180**, 1909-1925 (2008).

4. Burgess S, *et al.* Guidelines for performing Mendelian randomization investigations. *Wellcome Open Res* **4**, 186 (2019).
5. Min JL, *et al.* Genomic and phenotypic insights from an atlas of genetic effects on DNA methylation. *Nature genetics* **53**, 1311-1321 (2021).

Reviewers' Comments:

Reviewer #1:

Remarks to the Author:

The authors have addressed my concerns and I have no further comments.

Reviewer #2:

Remarks to the Author:

The authors have revised their manuscript in details according to reviewers' comments, and I am pleased with the revision and response, and no more questions/comments about the revised manuscript.

Reviewer #3:

Remarks to the Author:

The updated discussion has addressed all of my concerns. My compliments to the authors on an excellent study.